

# Exploring the Crucial Role of Atmospheric Carbonyl Compounds in Regional Ozone heavy Pollution: Insights from Intensive Field Observations and Observation-based modelling in the Chengdu Plain Urban Agglomeration, China

Jiemeng Bao[1,2], Xin Zhang[1,2], Zhenhai Wu[1], Li Zhou[3], Jun Qian[4], Qinwen Tan[5], Fumo Yang[3], Junhui Chen[6], Yunfeng Li[7], Hefan Liu[5], Liqun Deng[6], Hong Li[1*]

[1]Chinese Research Academy of Environmental Sciences, State Key Laboratory of Environmental Benchmarks and Risk Assessment, Beijing 100012, China

[2]School of Environmental Science and Engineering of Peking University, State Key Joint Laboratory of Environmental Simulation and Pollution Control, Joint Laboratory of Regional Pollution Control International Cooperation of the Ministry of Education, Beijing 100871, China

[3]College of Carbon Neutrality Future Technology, Sichuan University, Chengdu 610065, China

[4]Sichuan Radiation Environment Management and Monitoring Central Station, Chengdu 611139, China

[5]Chengdu Academy of Environmental Sciences, Chengdu 610046, China

[6]Sichuan Academy of Eco-Environmental Sciences, Chengdu 610042, China

[7]School of Mechanical Engineering, Beijing Institute of Petrochemical Technology, Beijing 102617,China

*Correspondence to*: Hong Li (lihong@craes.org.cn)

**Abstract.** Gaseous carbonyl compounds serve as crucial precursors and intermediates in atmospheric photochemical reactions, significantly contributing to ambient ozone formation. To investigate the impact of gaseous carbonyls on regional ozone pollution, simultaneous field observations and observation-based modelling of ambient carbonyls were conducted at nine sites within the Chengdu Plain Urban Agglomeration (CPUA), China during August 4-18, 2019, when three episodes of regional heavy ozone pollution occurred across eight cities within CPUA. Throughout the study, the total mixing ratios of 15 carbonyls ranged from 10.70 to 35.18 ppbv, in which formaldehyde (48.1%), acetone (19.9%), and acetaldehyde (17.5%) were most abundant within the CPUA. Ambient levels of carbonyls and ozone showed some positive correlations in space (especially pronounced around Chengdu in both northern and southern directions) and



in diurnal variations with higher concentrations of carbonyls during ozone pollution
episodes. Photochemical reactivity analysis emphasized the significant contributions of
carbonyls, especially formaldehyde and acetaldehyde, to ozone formation. The ozone
formation sensitivity for sites experiencing severe ozone pollution were classified as
VOCs-limited regime, while others were categorized as transitional regime. Local
primary emissions, mutual air transportation among cities within the CPUA and
photochemical secondary processes were recognized to contribute significantly to the
production or the contamination of carbonyls in ambient air, with alkenes and alkanes
being important secondary precursors of carbonyls. This study highlights the pivotal
role of carbonyls in heavy ozone pollution within the CPUA, China, providing valuable
scientific insights to guide the development of effective countermeasures for regional
ozone pollution control in the future.
**Keywords:** Gaseous Carbonyls; Ozone Heavy Pollution; Pollution Characteristics;
Atmospheric Photochemical Reactivity; Source Analysis; The Chengdu Plain Urban
Agglomeration, China
**1.  Introduction**
Atmospheric carbonyl compounds are pivotal in tropospheric chemistry, serving
as essential precursors to both ozone($O_3$) and secondary organic aerosols(SOA) (Guo
et al., 2004). Over the past two decades, severe air pollution in China has driven
substantial research efforts to understand the contributions of carbonyl compounds to
these environmental challenges. Studies have shown that photolysis of carbonyl
compounds is a major source of $RO_X$ radicals (Guenther et al., 2012; Y. Zhang et al.,
2016). These compounds can be photolyzed and react with OH radicals to form a large
number of $HO_2$ and $RO_2$ radicals, which increase the atmospheric oxidation capacity
and participate in the NOx photochemical cycle, leading to ozone formation (Y. Zhang
et al., 2016; Meng et al., 2017). Additionally, dialdehydes such as glyoxal and
methylglyoxal undergo heterogeneous reactions with aqueous particulate matter,
rapidly forming SOA (Lou et al., 2010; Xue et al., 2016; Yuan et al., 2012). Ambient





carbonyl compounds not only affect the environment but also pose direct health risks
to humans. They can harm ecosystems through deposition and adsorption processes
(Yang et al., 2018). They also pose direct health risks to humans, including sensitization,
carcinogenesis, and mutagenicity (Fuchs et al., 2017).

Significant progress has been made globally in understanding the concentrations

(Xue et al., 2013; Duan et al., 2012), diurnal variations(Shen et al., 2013; Fu et al.,
2008), and sources of carbonyl compounds(Pang and Mu, 2006; Rao et al., 2016). The
results highlight the severity and spatial-temporal variations of carbonyl pollution in
China. The results highlight severe and spatiotemporal variations of carbonyl pollution
in China. High levels are found mainly in the North China Plain, the Yangtze River
Delta, and the Pearl River Delta(Duan et al., 2008; Shao et al., 2009; Tan et al., 2018;
Wang et al., 2018; Xue et al., 2014, 2013; Yang et al., 2017). Urban areas show higher
carbonyl levels than suburban and rural areas due to human activities(Xue et al.,
2013).Despite many studies focusing on urban areas in China and comparing carbonyl
compound concentrations across different regions, there is a lack of comprehensive
analysis of atmospheric carbonyl compounds over larger areas, such as urban
agglomerations. In addition, most ground observations have been concentrated in fast-
developing regions, such as the NCP, YRD, and PRD. Existing research often
emphasizes overall VOCs rather than specific carbonyl compounds and their roles in
ozone pollution, leading to an incomplete understanding of the mechanisms by which
carbonyl compounds contribute to ozone formation and their regional differences.

Monitoring carbonyl compounds in the atmosphere is challenging due to their

typically low concentrations (ppt-ppb levels), necessitating highly sensitive analytical
methods. The diversity of carbonyl compounds, including multiple isomers, requires
highly selective analytical techniques for differentiation. Current measurement
technologies limit our understanding of the spatiotemporal distribution of carbonyl
compounds, affecting the accurate assessment of their environmental behavior, sources,
and transport. While previous studies have recognized the importance of carbonyl



compounds in ozone formation, detailed evaluations of their specific roles remain
insufficient.

Atmospheric carbonyl compounds originate from both primary and secondary

sources. Primary sources include the incomplete combustion of fossil fuels and biomass,
industrial emissions, emissions from the catering industry, and releases from plants.
Secondary sources arise from the atmospheric photochemical oxidation of VOCs
(Xue et al., 2013), particularly alkenes, aromatics, and isoprene, which typically
dominate the secondary formation of carbonyls. Existing source apportionment
methods, such as characteristic species ratio, source tracer proportion, multiple linear
regression, parameterization method based on photochemical age, and acceptor model,
struggle to distinguish between primary sources and secondary formation accurately.
The emission patterns of primary sources, particularly non-vehicle sources, are not well
understood. The source apportionment results elucidate the necessity to
comprehensively understand the secondary formation mechanisms of carbonyls.
Despite advancements in the study of atmospheric carbonyl compounds, significant
gaps remain in understanding their spatiotemporal distribution, source apportionment,
and contribution to ozone pollution. These gaps limit our comprehensive understanding
of the behavior of carbonyl compounds in the atmosphere, particularly in specific
regions and larger areas.

In this context, this study focuses on atmospheric carbonyl compounds and their

roles in photochemical pollution within the Chengdu Plain Urban Agglomeration
(CPUA) of China. The CPUA includes eight cities: Chengdu, Mianyang, Deyang,
Leshan, Meishan, Yaan, Suining, and Ziyang. This region has a developed economy
and a high degree of internationalization. The CPUA is located on the western edge of
the Sichuan Basin, surrounded by mountain ranges, which easily block airflow. The
unique climatic environment of the CPUA features low wind speeds year-round, high
frequency of static winds, short hours of sunshine, frequent winter inversions, and a
pronounced heat island effect in summer. These climatic characteristics significantly

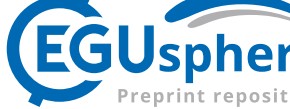

impact the variations in air pollutant concentrations, making the region prone to ozone
pollution in summer and haze pollution in winter. (Li et al., 2013; Hu et al., 2017; Zhang
et al., 2010). Although previous studies have shown that ozone formation in urban
Chengdu is primarily VOCs-limited (Tan et al., 2018), with aromatic hydrocarbons and
alkenes contributing significantly to ozone generation in summer (Xu et al., 2020),
these studies mainly focus on single cities and overall VOCs. There is limited
understanding of the distribution, sources, and specific roles of carbonyl compounds
across the entire CPUA and their contributions to regional ozone pollution and mutual
air transport mechanisms.

To address these research gaps, this study involves an intensive field observation

experiment conducted by the Sichuan Academy of Environmental Sciences, Peking
University, Sichuan University and Chinese Academy of Environmental Sciences.
Atmospheric carbonyl compounds were observed at nine sites in eight cities within the
CPUA for 15 days during a period of heavy ozone pollution in August 2019. Samples
were analyzed using 2,4-dinitrophenylhydrazine solid phase adsorption/high
performance liquid chromatography (HPLC). The study aims to characterize the
atmospheric carbonyl compounds in the CPUA, assess their influence on
photochemical pollution, identify key carbonyl compounds that may play crucial roles
in heavy ozone pollution in the CPUA, and evaluate the contribution of primary
emissions, air pollution transport, and secondary generation to key carbonyl compounds
through a combination of multivariate linear regression modeling and OBM. This
research aims to provide technical support for controlling carbonyl compounds
pollution in the CPUA and to reduce their contributions to ozone pollution.
**2. Materials and methods**
**2.1 Observation Sites Profile**

In this study, a total of 9 off-line sampling sites for atmospheric carbonyl

compounds were set up in 8 cities in the CPUA from August 4th to 18th, 2019(table S1).



Considering that this study focused on the pollution characterization of carbonyl
compounds in urban areas, one urban site was selected in each city. In addition, in order
to compare and study the pollution characteristics of carbonyl compounds in the
suburbs, a suburban site was set up in XJ County, Chengdu City. For the selection of
urban sites in each city, priority is given to those choices of set-up in the vicinity of the
state-controlled site, and the perimeter of the sites should be open, unobstructed and no
obvious pollution sources, with convenient transportation and power supply. The
distribution of specific sites is shown in Fig. 1.

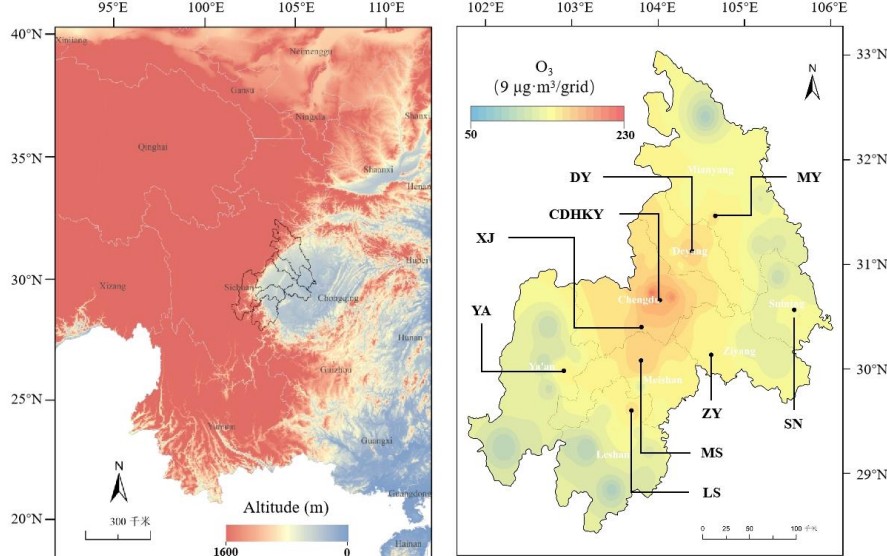

**Figure 1.** Sampling sites distribution.
**2.2 Samples Collection**
The sampling of atmospheric carbonyl compounds mainly referred to the TO-11A
standard of the United States Environmental Protection Agency (US EPA) and the
Chinese environmental protection standard HJ 683-2014 High Performance Liquid
Chromatography Method for the Determination of Atmospheric Carbonyl Compounds,
and the sampling was carried out by using silica gel sampling tubes (IC-DN3501 from
Tianjin Bonna-Agela) coated with DNPH (2,4-dinitrophenylhydrazine). In this study,
an automatic sampler for carbonyl compounds (Zhang et al., 2019) was used to



continuously collect atmospheric carbonyl compounds. From August 4th to 18th, 2019,
air samples were collected every 2 hours with a sampling flow rate of 0.8 L/min. In
addition, in order to prevent the impact of ozone and rainwater in the atmospheric air
on sample collection, a potassium iodide ozone removal column (KI 140 from Tianjin
Bonna-Agela) was installed and a water removal agent made by ourselves (Bao et al.,
2022; Wang et al., 2020) was added at the front end of the sample tube. Two blank
samples were collected before and after the sampling, and blank samples were also
collected for different batches of sampling tubes. The samples were frozen at -18°C and
analyzed within one month.
Atmospheric VOCs were sampled using SUMMA tanks, stainless steel tanks with
electropolished and silanized inner walls, manufactured by Entech in the United States,
with a sampling volume of 3.2 liters. The sampling was controlled by a constant current
integral sampler to sampling for an average of 1 hour. The sampling time was from
August 4th to 18th, 2019, and 2 VOCs samples were collected per day at each site (not
collected under special weather conditions such as rain), and each sample was collected
for 1 hour controlled by a cross-flow integration sampler. One sample was collected
from 8:00 to 9:00, and one sample was collected from 14:00 to 15:00, of which 6
samples were collected per day on August 11th, 12th and 16th (8:00-9:00, 10:00-1:00,
12:00-13:00, 14:00-15:00, 16:00-17:00, and 18:00-19:00).
**2.3 Samples Analysis**
The carbonyl compounds samples were qualitatively and quantitatively analyzed
by using High Performance Liquid Chromatography (HPLC) (LC-20AD, Shimadzu,
Japan) and an ultraviolet detector (SPD-20A, Shimadzu, Japan), mainly based on the
US EPA TO-11A standard and the Chinese HJ 683-2014 standard. The DNPH sampling
column after sampling was slowly eluted into a volumetric flask using acetonitrile
(chromatographically pure, Thermo Fisher Scientific China) to 5.0 mL. Then 1.5 mL
sample was taken into an HPLC sample bottle, and sealed and stored in a refrigerator
at <4 °C to complete the pre-treatment. Prior to sample analysis, a standard solution of



the concentration gradient was prepared using TO-11A standard solution (Supelco,
USA) and used as the external standard. The correlation coefficient ($R^2$) of the standard
curve was greater than 0.995. The limit of detection of the device was 0.56~5.57 ng/mL,
and the limit of quantification was 1.87~18.56 ng/mL (Table S2). Then 20 μL of the
pretreated sample was extracted through the autosampler and injected into the
HPLC/UV system, detected by a UV detector with a wavelength of 360 nm, qualified
by retention time value, quantified by peak area value, and the qualitative and
quantitative analysis data of carbonyl compounds were obtained after conversion. The
HPLC conditions referred to Chinese environmental protection standard HJ 683-2014:
binary gradient washing was performed using acetonitrile and water, 60% acetonitrile
was held for 20 mins, acetonitrile was increased linearly from 60% to 100% within 20-
30 mins, and acetonitrile was reduced to 60% again within 30-32 mins and held for 8
mins; the column oven was kept at 40 ℃.
The atmospheric VOCs were analyzed using the TO-14 and TO-15 methods
recommended by the US EPA, that is, frozen preconcentration coupled with gas
chromatography and mass chromatography. The sample was pre-concentrated by
Entech7100 system at a low temperature, then the VOCs components were quantified
by Agilent gas chromatography coupled with mass spectrometry instrument (GC-MS).
The concentrated samples were separated by gas chromatography and then entered
mass spectrometry for detection. A hydrogen flame ionization detector (FID) was used
to detect 5 substances: ethane, ethylene, acetylene, propane, and propylene. During the
sample analysis, four internal standard gases bromochloromethane, 1,4-
difluorobenzene, chlorobenzene-d5 and 4-bromofluorobenzene were used. With a
standard gas containing 118 substances such as PAMS, TO-15 and carbonyl compounds,
a multi-point calibration standard working curve was established using 6 concentration
gradients.
**2.4 Data Analysis**
**2.4.1 Ambient levels comparison**



According to the Technical Regulation on Ambient Air Quality Index (on trial),
National Environmental Protection Standard of the People's Republic of China HJ
633—2012, days with an ozone pollution index (IAQI) of 100 or higher during the
observation period were designated as pollution days, while days with an IAQI below
100 were considered clean days. This study compared the pollution characteristics of
carbonyl compounds between pollution days and clean days. Additionally, the
concentrations of formaldehyde, acetaldehyde, and acetone observed during the
summer of 2009-2013 in economically developed and industrialized areas such as
Beijing, Shanghai, and Guangzhou in China, as well as locations in South America
(Brazil), Asia (Thailand), Europe (France), and North America (United States), were
selected and compared.
**2.4.2 Ozone formation sensitivity inferring**
Previous studies have shown that the formaldehyde to $NO_2$ ratio (FNR) can be
used to determine the sensitivity of $O_3$-NOx-VOCs (Schroeder et al., 2017; Tonnesen
and Dennis, 2000; Vermeuel et al., 2019). Most studies used satellite remote sensing-
based FNR, but the FNR column concentration ratios inverted by satellite remote
sensing mainly represented the average photochemical of the troposphere, and the
concentration distributions of HCHO and $NO_2$ in the vertical direction were
inconsistent (Hong et al., 2022; Schroeder et al., 2017). So, there is a large uncertainty
to develop ground-level ozone pollution prevention and control measures. In this study,
sensitivity analysis of ground-level ozone formation was carried out based on the ratio
of ground-level HCHO to $NO_2$ during the observation period at the 9 sites of 8 cities in
the CPUA. FNR < 0.55±0.16 and FNR > 1.0±0.3 were defined to VOCs-limited and
NOx-limited, respectively, and FNR ratio ranged from 0.55±0.16 to 1.0±0.3 defined to
$NO_X$ and VOCs co-limited (Liu et al., 2021; Zhang et al., 2022).
**2.4.3 Secondary formation mechanism investigation**

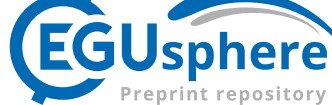

**(1) Atmospheric chemical reactivity**
In this study, the contribution of atmospheric chemical reactivity of carbonyl
compounds to ozone formation was evaluated using the OH free radical consumption
rate ($L_{OH}$) and ozone formation potential (OFP):
$$L_{OH} = [OVOC]_i \times K_i(OH) \qquad (1)$$
Where, $[OVOC]_i$ was the observed concentration of the $i^{th}$ (i=1 to n) carbonyl
compound, in molecule/cm$^3$; $K_i(OH)$ was the rate constants of the $i^{th}$ carbonyl
compound reacting with OH radicals, in cm$^3$/(molecule·s); the selected $K_i(OH)$ values
were from literature (Atkinson and Arey, 2003).
$$OFP = MIR_i \times [OVOC]_i \qquad (2)$$
Where, MIR was the maximum incremental reactivity of the $i^{th}$ carbonyl
compound, and the MIR values of each species were from California Code of
Regulations (https://govt.westlaw.com); $[OVOC]_i$ was the mass concentration of the $i^{th}$
carbonyl compound, in μg/m$^3$.
**(2) Observation-based model (OBM)**
The relative incremental activity (RIR) was calculated by assuming that the
concentration of a given carbonyl compound precursor decreased by a certain
proportion could cause the change of the concentration of the carbonyl compound, so
as to further judge the effect of VOCs on the formation of carbonyl compounds.
Combining the concentrations and activity levels of 15 carbonyl compounds during the
observation period, this study focused on formaldehyde, acetaldehyde, and acetone as
the primary research targets. The impacts of various AVOCs(anthropogenic VOCs),
including alkanes, alkenes, alkynes, and aromatic hydrocarbons, as well as BVOCs
(biogenic VOCs) like isoprene, on the formation of formaldehyde, acetaldehyde,
and acetone were assessed using observation-based OBM classification. Specific
species of anthropogenic source VOCs (alkanes, alkenes, alkynes, and aromatic
hydrocarbons) and biogenic VOCs (isoprene) are detailed in Table S3.





VOCs observations, conventional gases (NO$_2$, CO and SO$_2$) and meteorological
parameters (temperature, relative humidity and pressure) were imputed into the model.
It was assumed that the pollutants are well mixed. Under the constraints of the measured
hourly concentration data of pollutants, the atmospheric chemical process was
simulated to obtain the source-effect relationship of the measured pollutants. By
assuming the reduction of the source effect, the RIRs of different carbonyl compounds
precursors were calculated, and the sensitivities of carbonyl compounds to different
pollutants were obtained, and then the secondary formation mechanism of carbonyl
compounds was determined. The formula to calculate the RIR is as follows:
$$RIR(X) = \left[\frac{\Delta P_Y(X)/P_Y(X)}{\Delta S(X)/S(X)}\right] \qquad (3)$$
$$P_Y = Y_{\text{net formation}} - Y_{\text{net consumption}} \qquad (4)$$
Where X was a specific species; $P_Y(X)$ was the net formation rate of species y;
S(X) was the total amount of emissions of species X in a certain period, i.e., the source
effect of species X. $\Delta S(X)$ was the change in total emissions of X caused by the
hypothetical change in source effect, $\Delta P_Y(X)$ was the change in $P_Y(X)$ after the change
in source effect S(X), and RIR(X) was the relative incremental reactivity of species X.
The species Y in this study were formaldehyde, acetaldehyde and acetone, respectively,
and pollutant X was reduced by 20%.
The absolute RIR of the precursor reflects the sensitivity of carbonyl compounds
formation to the precursor. The higher the absolute RIR, the more sensitive the carbonyl
compounds formation to the precursor. A positive RIR value indicates that reducing the
species can reduce the formation rate of species Y, and a negative RIR value indicates
that reducing the species can increase the formation rate of species Y.
**2.4.4 Sources Analysis**
**(1) Multi-linear regression model**
There is a good correlation between concentrations of compounds of the same or



similar source in the atmosphere. Based on this property, it was assumed that the
primary and secondary sources of carbonyl compounds were linearly correlated with
the selected tracers, and then a quantitative source model was established by multiple
linear statistical regression analysis (Kanjanasiranont et al., 2016a; Li et al., 2010; Ling
et al., 2017; Luecken et al., 2012; Lui et al., 2017; Wang et al., 2017). In general, CO is
the marker product of typical anthropogenic combustion source emissions, mainly from
vehicle exhaust emissions and coal combustion. Ozone, as an indicator of
photochemical smog, is a typical secondary formation pollutant. In this study, CO and
ozone were selected as the tracers of primary source and secondary source of carbonyl
compounds, respectively. The formula is as follows:
$$[carbonyl] = \beta_0 + \beta_1[CO] + \beta_2[O_3] \tag{6}$$

Where [carbonyl], [CO] and [$O_3$] represented the observed mixing ratios of
carbonyl compounds, CO and ozone, respectively, in ppbv. $\beta_0$, $\beta_1$ and $\beta_2$ were
coefficients obtained by multiple linear regression fitting model, in ppbv/ppbv. $\beta_0$
represented the background concentration of a given carbonyl compound, $\beta_1$
represented the emission ratio of the carbonyl compound relative to CO. $\beta_1[CO]$ and
$\beta_2[O_3]$ represented the concentrations of carbonyl compound in primary emission and
secondary formation, respectively, in ppbv.
In addition, the relative contribution of primary emissions, secondary formation
and background concentrations of carbonyl compounds can be calculated using the
following formula:
$$P_{primary} = \frac{\beta_1[CO]_i}{(\beta_0 + \beta_1[CO]_i + \beta_2[O_3]_i)} \times 100\% \tag{7}$$

$$P_{secondary} = \frac{\beta_2[O_3]_i}{(\beta_0 + \beta_1[CO]_i + \beta_2[O_3]_i)} \times 100\% \tag{8}$$

$$P_{background} = \frac{\beta_0}{(\beta_0 + \beta_1[CO]_i + \beta_2[O_3]_i)} \times 100\% \tag{9}$$

Where, $P_{primary}$ represented the contribution of the primary emission of a given
carbonyl compound, %; $P_{secondary}$ represented the contribution of the secondary
formation of the carbonyl compound species, %; $P_{background}$ represented the contribution





of the carbonyl compounds species from sources other than primary emissions and
secondary formation, %.
**(2) Backward trajectory model**
The effects of long-distance air mass transport on the pollution of carbonyl
compounds in the CPUA were studied using MeteoInfo software and TrajStat plug-in.
In this model, meteorological data were relevant meteorological data from the global
date           assimilation           system           (GDAS)           database
([ftp://arlftp.arlhq.noaa.gov/pub/archives/gdasl](ftp://arlftp.arlhq.noaa.gov/pub/archives/gdasl)). A trajectory simulation height of 500
m was selected. The duration of backward trajectory was 48 h. The daily start time was
00:00 UTC. The analog frequency was 2 h. The backward trajectory diagram was
calculated. Meanwhile, the clustering method in TrajStat software and the Euclidean
distance algorithm were used to cluster the airflow trajectory to the CPUA. And then
the statistical analysis was carried out in combination with the corresponding pollutant
mass concentration characteristics.
**3. Results and Discussion**
**3.1 Overview of air quality during observation period**
Due to the influence of cooling and precipitation caused by cold air intrusion, the
early observation period (from August 4th to 6th, 2019) in the Chengdu Plain Urban
Agglomeration (CPUA) experienced slightly lower temperatures (25.1°C) and higher
humidity (87.6%). These conditions were not conducive to ozone formation. However,
as temperatures rose and humidity decreased thereafter, favorable conditions for ozone
generation emerged, leading to heavy and persistent regional ozone pollution in the
CPUA. By August 12th, the mean temperature had gradually increased to 29.1°C, while
it averaged 27.7°C from August 13th to 14th. During this time, cumulative precipitation
reached 975 mm, resulting in temporary alleviation of ozone pollution. Subsequently,
temperatures rose again from August 15th to 18th, with the mean temperature persisting



above 28.4°C for several days, accompanied by a decrease in humidity to a minimum
of 64.8% on August 17th. Overall, during the observation period (from August 4th, 2019,
0:00 to August 18th, 2019, 24:00), three episodes of severe ozone pollution occurred,
namely EP1 (August 7th to 9th), EP2 (August 10th to 13th), and EP3 (August 15th to 18th),
as depicted in Fig. 2.

Fig.3 illustrates the temporal and spatial variations of ozone and $NO_2$

concentrations, as well as temperature and humidity at each site during the observation
period. After observing the spatial distribution of ozone concentration during EP1, it's
evident that the severity of pollution reached heavily polluted levels, with Chengdu
recording an $O_3$-8h concentration of 297 μg/m$^3$ on August 7th. This distribution
demonstrated a radial decrease from Chengdu to the surrounding areas. However, the
subsequent episodes, EP2 and EP3, exhibited even broader ranges of ozone pollution
and more pronounced spatial movements. During the early stages of EP2 and EP3 (from
August 10th to 11th and from August 14th to 15th, respectively), high ozone
concentrations were observed in the Chengdu-Deyang-Mianyang region. In the middle
stages (August 12th and from August 16th to 17th, respectively), influenced by northerly
airflow, regions with high ozone concentrations expanded to the central (Meishan,
Ziyang, and Suining) and southwestern (Leshan and Ya'an) parts of the CPUA. In the
later stages (August 13th and August 18th), under the influence of northwesterly airflow,
regions with high ozone concentrations (Meishan and Leshan) moved southward again,
while ozone pollution in other areas of the CPUA gradually weakened. On August 11th
to 12th and August 16th to 17th, ozone concentrations in the eight cities of the CPUA
reached light pollution levels or higher, with the heaviest pollution recorded on August
12th. Specifically, Deyang, Mianyang, Suining, and Meishan reached moderate
pollution levels, while Chengdu reached heavy pollution with a concentration of 324
μg/m$^3$.



**Figure 2.** Overview of air quality at each site during the observation period. The gray shaded parts




respectively represent the three heavy ozone pollution episodes (EP1,EP2,EP3).

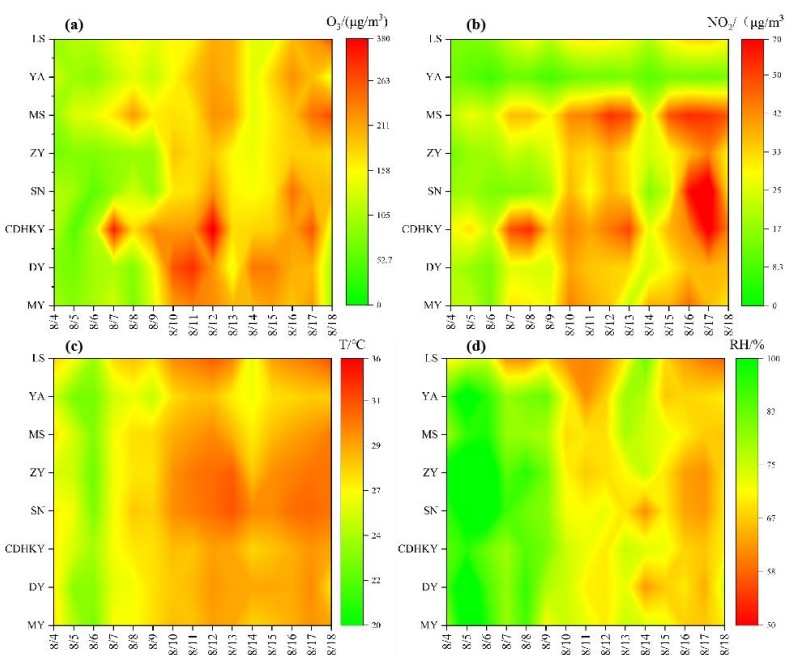

**Figure 3.** Temporal and spatial variations of (a) ozone concentration, (b) NO$_2$ concentration, (c)

temperature and (d) humidity in the CPUA during the observation period.

**3.2 Comparative characterization of carbonyl compounds**
**3.2.1 Ambient levels**

During the observation period, we utilized 2,4dinitrophenylhydrazine (DNPH)

cartridge and high-performance liquid chromatography (HPLC) analysis technique to
quantify 15 carbonyl compounds. The concentrations and relative proportions of these
compounds are summarized in Table 1. The average concentration of the 15 carbonyl
species in the CPUA was 17.35 ± 5.31 ppb. Overall, areas with elevated concentrations
of carbonyl compounds were primarily concentrated in and around Chengdu in both
northern and southern directions. MY site, located to the north of Chengdu, exhibited
the highest concentration of carbonyl compounds (35.18 ± 13.37 ppb), while YA site,
situated southwest of Chengdu, showed the lowest concentration (10.70 ± 4.16 ppb).



**Table 1.** Daily mean mixing ratio of carbonyl compounds at each site in the CPUA during the
observation period (ppbv)

| Carbonyls | MY | DY | CDHKY | XJ | SN | ZY | MS | YA | LS |
|---|---|---|---|---|---|---|---|---|---|
| formaldehyde | 12.82±6.52 | 6.06±2.82 | 10.09±4.21 | 8.87±4.39 | 6.98±3.56 | 5.84±2.69 | 8.47±4.15 | 6.36±2.40 | 6.55±3.35 |
| acetaldehyde | 16.65±7.38 | 1.54±0.77 | 3.65±2.15 | 2.33±1.07 | 2.62±1.74 | 1.40±0.61 | 3.24±1.60 | 0.88±0.68 | 1.63±1.32 |
| acetone | 4.36±1.70 | 2.80±1.19 | 4.51±2.25 | 3.70±1.21 | 3.14±1.70 | 3.23±1.73 | 2.15±1.14 | 2.18±1.08 | 2.91±1.63 |
| propionaldehyde | 0.41±0.22 | 0.24±0.14 | 0.39±0.27 | 0.39±0.17 | 0.34±0.22 | 0.28±0.14 | 0.41±0.18 | 0.20±0.15 | 0.31±0.16 |
| crotoraldehyde | 0.20±0.21 | 0.10±0.11 | 0.23±0.34 | 0.05±0.07 | 0.23±0.08 | 0.19±0.27 | 0.15±0.21 | 0.36±0.24 | 0.12±0.24 |
| butyaldehyde | 0.22±0.48 | 0.22±0.28 | 0.40±0.57 | 0.94±1.67 | 0.26±0.18 | 0.06±0.18 | 0.44±0.46 | 0.25±0.16 | 0.02±0.06 |
| benzaldehyde | 0.00±0.04 | 0.02±0.06 | 0.04±0.11 | 0.21±0.20 | 0.08±0.10 | 0.00±0.01 | 0.00±0.01 | 0.00±0.00 | 0.01±0.04 |
| isovaleraldehyde | 0.01±0.14 | 0.03±0.09 | 0.08±0.14 | 0.08±0.13 | 0.05±0.10 | 0.01±0.05 | 0.68±0.42 | 0.04±0.07 | 0.06±0.12 |
| valeraldehyde | 0.00±0.00 | 0.25±0.09 | 0.30±0.59 | 0.63±0.36 | 0.85±0.65 | 0.00±0.00 | 0.00±0.00 | 0.00±0.02 | 0.77±0.47 |
| o-Tolualdehyde | 0.46±0.52 | 0.36±0.29 | 0.45±0.19 | 0.00±0.00 | 0.00±0.00 | 0.23±0.17 | 0.43±0.33 | 0.18±0.22 | 0.16±0.17 |
| m-Tolualdehyde | 0.00±0.02 | 0.04±0.10 | 0.04±0.09 | 0.17±0.17 | 0.30±0.13 | 0.00±0.03 | 0.00±0.02 | 0.00±0.02 | 0.01±0.05 |
| p-Tolualdehyde | 0.00±0.00 | 0.01±0.05 | 0.01±0.04 | 0.00±0.00 | 0.00±0.00 | 0.00±0.00 | 0.01±0.04 | 0.00±0.02 | 0.00±0.02 |
| hexaldehyde | 0.00±0.01 | 0.34±0.25 | 0.41±0.69 | 0.57±0.47 | 0.95±0.65 | 0.02±0.18 | 0.78±0.58 | 0.00±0.01 | 0.10±0.32 |
| 2,5-diemthybenzaldehyde | 0.01±0.03 | 0.00±0.01 | 0.00±0.01 | 0.05±0.12 | 0.00±0.00 | 0.00±0.01 | 0.01±0.02 | 0.00±0.01 | 0.00±0.01 |
| MACR | 0.03±0.20 | 0.14±0.17 | 0.26±0.34 | 1.05±1.10 | 0.26±0.21 | 0.19±0.16 | 0.42±0.36 | 0.24±0.22 | 0.81±0.88 |
| **Sum** | 35.18±13.37 | 12.16±4.84 | 20.84±8.85 | 19.04±8.1 | 16.05±7.73 | 11.47±4.89 | 17.19±7.61 | 10.70±4.16 | 13.46±6.12 |






397   Fig.S1 illustrates the relationship between ozone concentration and carbonyl

398 compounds concentration at each site during the observation period. It is evident that

399 the spatial distribution of carbonyl compound concentrations is similar to that of ozone

400 concentration. Regions with severe ozone pollution tend to exhibit higher

401 concentrations of carbonyl compounds. The variation in carbonyl compound

402 concentrations is primarily attributed to anthropogenic emissions and prevailing

403 summer wind directions in the CPUA. Chengdu is the most economically developed

404 city in the CPUA, with notably higher GDP and industrial production values than other

405 regions. Chengdu's major industries include coal-fired power plants, chemical plants,

406 metallurgy and building materials plants, and high concentrations of carbonyls were

407 observed in here. The unique basin climate of the CPUA, characterized by intense

408 sunlight and stable atmospheric conditions, facilitates the accumulation of pollutants.

409 Large amount of industrial emissions and strong photochemical reaction contributes to

410 ozone pollution. Additionally, during the summer, prevailing northerly winds in the

411 CPUA facilitate the downwind transport of pollutants from upwind sources, leading to

412 regional pollution. It is noteworthy that the concentration of carbonyl compounds at the

413 MY site significantly exceeds that at the CDHKY site. MY, with its industrial roots,

414 consistently maintains its position as the second-highest GDP contributor in Sichuan

415 Province. The electronics information industry stands as Mianyang's primary economic

416 driver, constituting approximately half of the city's total output value. Studies

417 investigating the volatile organic compound (VOC) source profile in Chengdu(Zhou et

418 al., 2021) reveal that ethanol and carbonyls predominantly characterize electronics

419 manufacturing emissions.

420 **3.2.2 Compositional characteristics**

421   According to the composition characteristics of 15 carbonyl compounds in the

422 ambient air of each city during the observation period(Table S4). Formaldehyde was





the most abundant specie found in these sites followed by acetone and acetaldehyde,
which is widely observed in previous studies. The concentration ratios of formaldehyde,
acetone, and acetaldehyde across different sites ranged from 36.4% to 59.4% (average
48.1%), 12.4% to 28.1% (average 19.9%), and 8.2% to 47.3% (average 17.5%),
respectively. In this study, the total concentrations of formaldehyde, acetaldehyde, and
acetone(FAT)account for over 78% of the total carbonyls concentrations. At the MY
and ZY sites, this proportion even exceeded 90%. It is noteworthy that isobutyraldehyde
(MACR) ranks fourth in the volume concentration of 15 carbonyls in the ambient air
surrounding XJ, accounting for 5.3%. MACR, a characteristic product of isoprene
photooxidation from biogenic sources, possibly originates from the abundant
vegetation surrounding XJ. It reflects the period's relatively active photochemical
reactions, with substantial contributions from secondary formation to the carbonyls
composition.

The observed levels of FAT in different areas were influenced by various factors

including sampling period, geographic location, meteorological conditions, chemical
removal, and source emissions(Z. Zhang et al., 2016). Despite these influences,
comparisons remain valuable in providing an overview of ambient carbonyl levels in
the CPUA. During the summer of 2010, a national wide survey of ambient
monocarbonyl compounds were conducted simultaneously in nine sites (Ho et al.,
2015)found that the total FAT concentration was highest in Chengdu (14.96 ppb),
followed by Beijing (11.83 ppb), and Wuhan (11.70 ppb). Beijing, as the capital of
China, and Wuhan, being one of the top ten most populous cities in China, played
significant roles in this comparison. In our study, the CDHKY site within CPUA
exhibited the highest FAT concentration, with values of 18.25 ppb, surpassing those
recorded in 2010. Furthermore, the total FAT concentrations observed at the CPUA and
XJ sites, with values of 14.99 ppb and 14.90 ppb respectively in our study, closely
resemble those reported in August 2010 in Chengdu. This suggests that elevated
concentrations of carbonyl compounds in Chengdu have been a longstanding issue on



a national scale. Comparing our findings to international studies, the FAT
concentrations at the CDHKY site were lower than those reported in Rio De Janeiro,
Brazil(da Silva et al., 2016), during July to October 2013 (35.43 ppb), but higher than
those in Bangkok, Thailand(Kanjanasiranont et al., 2016b), Orleans, France(Jiang et al.,
2016), and the United States(Murillo et al., 2012), with values of 9.05 ppb, 6.12 ppb,
and 5.76 ppb, respectively.
**3.3 Temporal variations of carbonyl compounds**
The diurnal variation of the total mixing ratio of ambient carbonyl compounds and
ozone concentration around each site in the CPUA during the observation period is
shown in Fig. 4. According to the observation results, the diurnal trend of ozone
concentration at each site showed a "unimodal" variation characteristic, that was, it
gradually increased from the morning to the peak of one day at noon, and then decreased.
The diurnal variation of the total mixing ratio of carbonyl compounds at each site
generally showed a characteristic of high during the daytime and low at night. The
concentration of carbonyl compounds during the day (6:00-16:00) was 48.8% higher
than that at night (18:00-4:00) at the XJ site. This indicated that the concentration of
carbonyl compounds increased by photochemical production during the daytime. The
diurnal variation characteristics of each site were different. For example, the diurnal
variation characteristics of carbonyl compounds concentration at CDHKY, XJ and SN
sites were consistent with those of ozone. The diurnal variation of carbonyl compounds
concentrations at other sites showed "double peaks", peaking at 10:00-12:00 and 18:00-
20:00, respectively. The concentrations of carbonyl compounds at night were also
higher at MY, DY and LS sites. The diurnal minimum values of the total concentration
of carbonyl compounds and ozone concentration appeared at similar time, usually at
4:00 a.m. or 6:00 a.m. The first peak of the total mixing ratio of carbonyl compounds
occurred earlier than the maximum ozone concentration of the day. The first peak of
the total mixing ratio of carbonyl compounds mostly occurred between 10:00 and 12:00.
And the maximum ozone concentration mostly occurs between 14:00 and 16:00. This



was related to the fact that carbonyl compounds were important precursors of ozone.

In general, the diurnal variation of the total concentration of carbonyl

compounds on pollution days and clean days was high during the daytime and low at
night. The total mixing ratio of carbonyl compounds on pollution days was 22.8%-
66.2% higher than that on clean days. At the same time, the increase of concentration
of carbonyl compounds during the daytime on pollution days was higher than that on
clean days. This suggested that the increase in the concentration of carbonyl
compounds during the daytime contributed to ozone pollution.

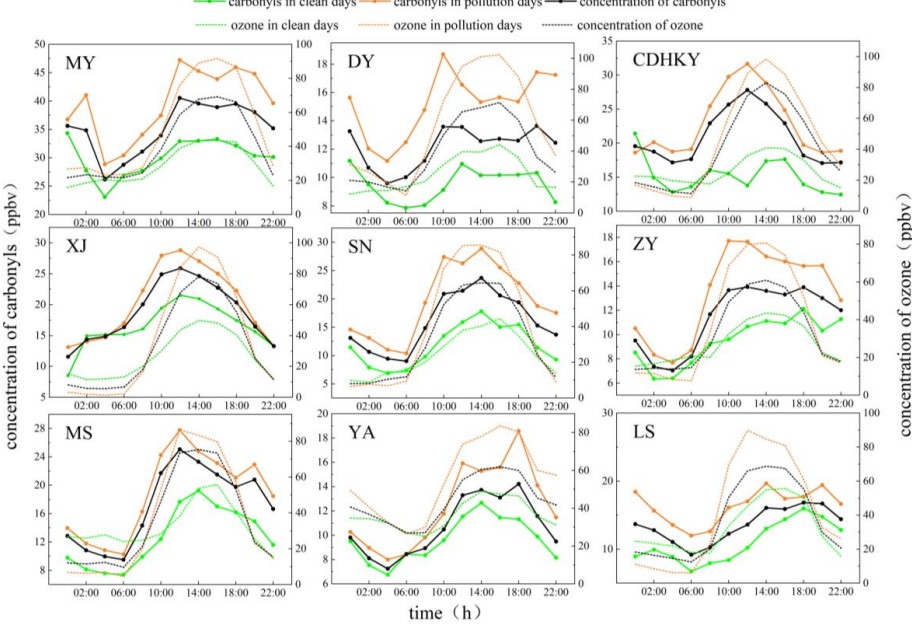

**Figure 4.** Diurnal variations of carbonyl compounds and ozone concentrations at each site in the

CPUA during the observation period

The diurnal variation of the mixing ratio of ambient carbonyl compounds on

weekdays and weekends in the eight cities of the CPUA is shown in Fig. S2. The total
concentration of carbonyl compounds at each site on weekends was higher than that on
weekdays, and the increase in carbonyl compounds at 0:00 (36.3%), 10:00 (16.3%) and
18:00-22:00 (17.6%) on weekends was higher than that on weekdays. Except for the
XJ site, the increase in the concentration of carbonyl compounds at 0:00 on weekends



was significantly higher than that on weekdays, which was mainly related to the
increase of acetaldehyde, propionaldehyde and acetone on weekends. At 10:00, the
higher increase at DY, CDHKY and SN sites was mainly related to the increase of
propionaldehyde, acetaldehyde and formaldehyde concentrations. From 18:00 to 22:00,
the higher increase at DY and YA sites was mainly related to the increase in the
concentrations of propionaldehyde, acetone and acetaldehyde. Acetaldehyde, acetone
and propionaldehyde were mainly from vehicle exhaust. In particular, when ethanol
gasoline and biodiesel were used as alternative fuels, the content of acetaldehyde and
acetone in the exhaust gas would be significantly increased. Therefore, the increase in
the concentration of carbonyl compounds on weekends might be related to the increase
in traffic at 10:00 and at night. In addition, the peak concentration of carbonyl
compounds on weekends (10:00) was earlier than that on weekdays (12:00-14:00) at
CDHKY, XJ and SN sites, and the diurnal trend of carbonyl compounds concentrations
on weekdays and weekends had little difference at other sites.
**3.4 Atmospheric photochemical reactivity of carbonyl compounds**

During the observation period, the total OH radical consumption rate ($L_{OH}$) and

total ozone formation potential (OFP) of the 15 carbonyl compounds at each site are
depicted in Fig.5. The ranking of total $L_{OH}$ and total OFP at each site is consistent,
except for the YA and ZY sites with lower concentrations of carbonyl compounds,
where the atmospheric photochemical reactivity ranking also aligns with the
concentration. Among all sites, the MY and CD sites display the highest reactivity,
while the YA and ZY sites exhibit the lowest reactivity. Contrasting the $L_{OH}$ and OFP
during clean and polluted periods reveals higher values during ozone pollution periods
than clean days. $L_{OH}$ and OFP during different pollution periods show a strong positive
correlation with the severity of ozone pollution; the heavier the ozone pollution, the
higher the $L_{OH}$ and OFP at the sites. Regardless of clean or polluted periods, the $L_{OH}$
and OFP at the MY site are higher than other sites. However, despite this, the average
ozone concentration at the MY site ranks lower among the nine sites observed. This



might be associated with higher concentrations of aldehyde compounds at the MY site.
During the observation period, carbonyl compounds significantly contributed to
ozone formation. The contributions to total VOCs (alkanes, alkenes, alkynes, aromatics,
and carbonyl compounds) OFP at the MY, SN, ZY, YA, and LS sites ranged from 19.5%
to 48.6%. Formaldehyde and acetaldehyde were identified as the most reactive species
in the atmosphere, surpassing other carbonyl compounds in reactivity due to their
higher concentrations and inherent reactivity, especially formaldehyde. However,
acetone exhibited high inertness and a prolonged atmospheric lifetime, leading to its
accumulation in ambient air with concentrations higher than other carbonyl compounds
except for formaldehyde and acetaldehyde. Thus, despite its elevated concentration,
acetone's reactivity remained relatively low.

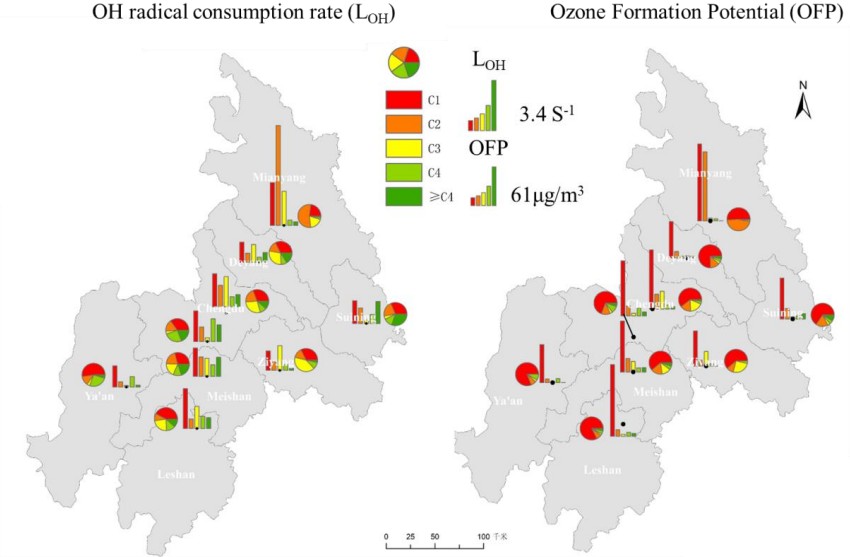


**Figure 5.** $L_{OH}$ and OFP of carbonyl compounds at each site in the CPUA during the observation
period
**3.5 Sensitivity analysis of ozone formation based on formaldehyde to NO$_2$ ratio (FNR)**
The change of O$_3$ formation sensitivity of each site in the CPUA during the
observation period is shown in Fig.3.9. As can be seen from the Fig. 6, most sites remain





in the VOCs-limited regime during the cleaning period and EP1 to EP3. Economically
developed city such as Chengdu, Meishan, with high levels of formaldehyde and $NO_2$,
remain in the VOCs-limited regime. Ya'an as a city with the lowest GDP ranking in the
CPUA, with low levels of formaldehyde and $NO_2$, remain in the transitional regime.

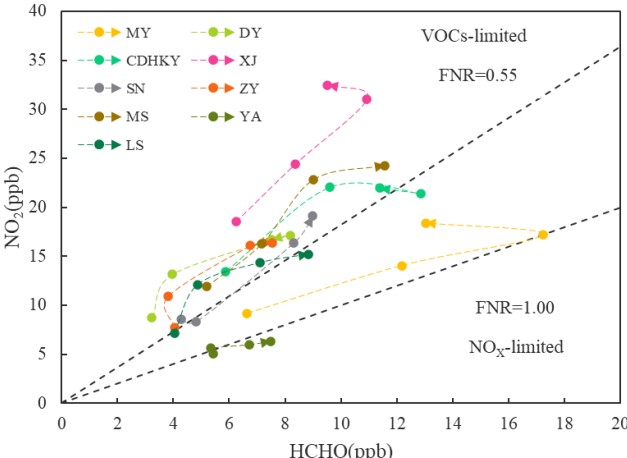


**Figure 6.** The change of $O_3$ formation sensitivity of each site in the CPUA during the observation
period. The arrows represent time step from clean period to EP1 to EP2 to EP3.
The daily variation of $O_3$ formation sensitivity and ozone concentration at each
site in the CPUA during the observation period is shown in Fig. S5. The mean FNR of
each site ranged from 0.48 to 1.29 during the observation period. The FNRs were lower
than 0.55±0.16 at XJ, DY, ZY, CDHKY, and MS, and higher than 1.0 at LS, SN, YA and
MY. At the same time, the mean ozone concentration at each site was between 138 and
192 μg/m³. The mean ozone concentration in XJ, DY, CDHKY and MS was 166-192
μg/m³, it was 150-164 μg/m³ in LS, SN, YA and MY. Therefore, it could be seen that
most of the sites with high mean ozone concentrations during the observation period,
like CDHKY, XJ, MS and Deyan sites, were in the VOCs-limited regime, and most of
the stations with low mean ozone concentrations during the observation period such as
YA, SN, MY and LS were in the transitional regime. It was worth noting that the mean
ozone concentration at ZY site (only 138 μg/m³) during the observation period was
much lower than that of other sites, but most of the ZY site was in VOCs-limited regime,



which was mainly related to the low concentration of formaldehyde. In addition, the
FNR value of the MY site was also relatively high, which was mainly caused by the
high concentration of formaldehyde.
Based on the ratio of formaldehyde to $NO_2$ mixing ratio, most sites remain in the
VOCs-limited regime during the observation period. And the sites with heavy ozone
pollution were in the VOCs-limited regime, and the sites with light ozone pollution
were in the transitional regime. Photochemical reactivity ($L_{OH}$ and OFP) analysis
showed that formaldehyde and acetaldehyde contributed significantly to the
enhancement of atmospheric oxidation and ozone formation potential. Therefore, when
heavy ozone pollution occurs in the CPUA, special attention should be paid to the
control of VOCs, especially formaldehyde and acetaldehyde in carbonyl compounds,
under the coordinated control of NOx and VOCs. Overall, this study reveals the
important contribution of carbonyl compounds to ozone pollution in the CPUA, and
provides scientific support for the establishment of ozone pollution prevention and
control measures.
**3.6 Source Analysis of carbonyl compounds**
**3.6.1 Quantitative source analysis of key carbonyl compounds**
The table S7 provides a summary of the background and primary emissions
concentrations of formaldehyde, acetaldehyde, and acetone at nine sites across the eight
cities of the CPUA, along with the proportion of secondary formation contributing to
their concentrations. Background concentrations and primary emissions of
formaldehyde, acetaldehyde, and acetone ranged from 50% to 80%, 46% to 83%, and
45% to 78%, respectively. Secondary formation accounted for 20% to 50%, 17% to
54%, and 22% to 55% of their concentrations, respectively. Notably, in SN and YA, the
secondary formation of formaldehyde contributed half of the observed concentration,
indicating it as the predominant source, while acetaldehyde's secondary formation also
prevailed in these sites. Conversely, acetone, with lower reactivity, primarily originated





from background concentrations and primary emissions at other sites except YA.
Moreover, background concentrations and primary emissions were identified as the
main contributors to carbonyl compounds in XJ and LS.
Fig.7 illustrates the secondary formation concentrations of formaldehyde,
acetaldehyde, and acetone at each site in the CPUA under both clean and polluted
conditions. Under polluted conditions, the secondary concentrations of formaldehyde,
acetaldehyde, and acetone exceeded those in clean conditions by 52.4%, 80.3%, and
58.5%, respectively. The most significant increases in secondary concentrations were
observed at the SN site, while relatively smaller increases were observed at LS and XJ.

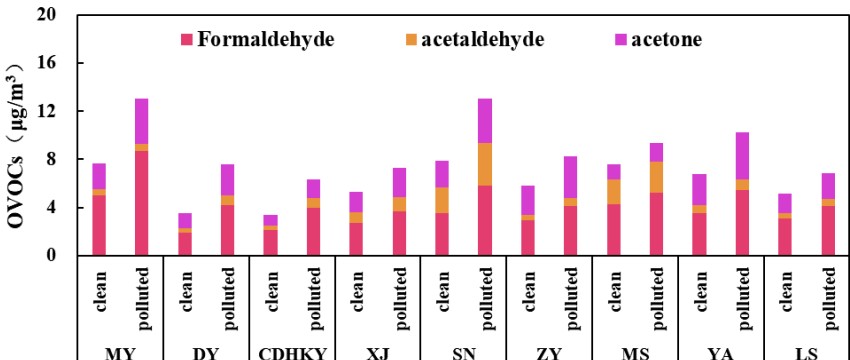


**Figure 7.** Concentrations of formaldehyde, acetaldehyde and acetone in secondary formation
under different pollution conditions at each site in the CPUA during the observation period
**3.6.2 Investigation of secondary formation mechanism of key carbonyl compounds**
The effects of anthropogenic source VOCs and plant source VOCs on the
formation of formaldehyde, acetaldehyde and acetone at MY, SN, ZY,YA and LS sites
were researched during a regional ozone pollution period when all 8 cities of the CPUA
had mild or above ozone pollution (August 11[th], 12[th] and 16[th]) (Fig.8). Overall, the
sensitivities of different anthropogenic source and plant source VOCs to formaldehyde,
acetaldehyde and acetone was consistent among sites. For formaldehyde, reducing
alkenes in anthropogenic source VOCs and plant VOCs was the most effective way to
control formaldehyde concentration, while reducing alkenes in anthropogenic source




VOCs was also beneficial to reduce the formation of acetaldehyde. For acetone with
low reactivity, the alkanes in anthropogenic source VOCs were the most sensitive to
the formation of acetone, followed by alkenes and BVOCs. Only the RIR value of
alkanes were greater than zero, and the RIR values of both alkenes and BVOCs were
less than zero, indicating that reducing alkanes could reduce the formation of acetone,
while reducing alkenes and BVOCs was not conducive to acetone concentration control.

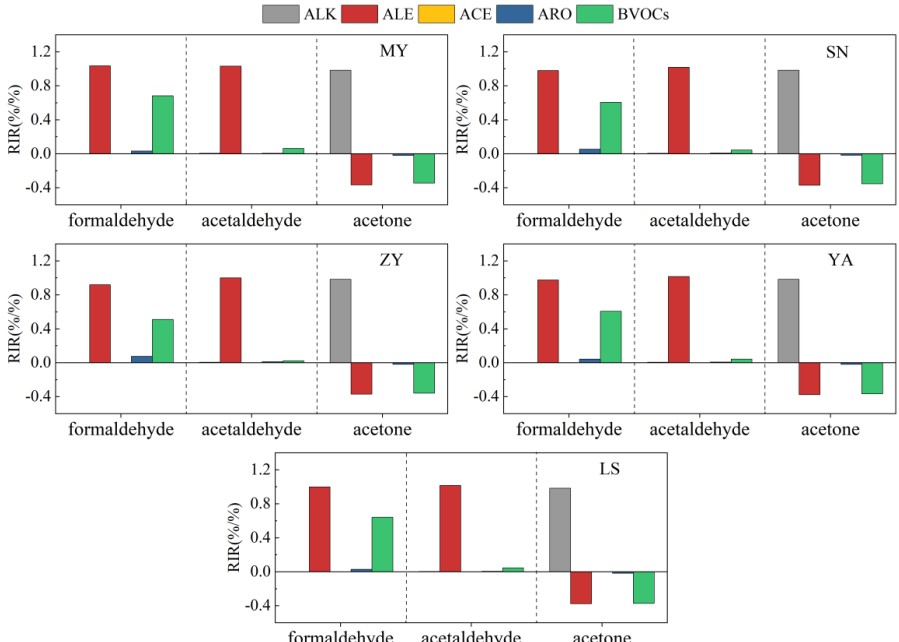


**Figure 8.** Mean RIRs of formaldehyde, acetaldehyde and acetone to different anthropogenic
source VOCs and biogenic source VOCs at MY, SN, ZY,YA and LS sites on August 11[th], 12[th] and

16[th]

**3.6.3 Influence of regional transportation contribution**

The TrajStat trajectory model was used to calculate and cluster the 24-hour

backward trajectories of air quality at the sampling sites. The backward trajectory
during sampling is shown in Fig.S6. During the observation period, the pollution of
carbonyl compounds in the cities of the CPUA was affected by the mutual transport
among cities in Sichuan Province, especially along the MY-DY-CDHKY route. In



addition, the surrounding provinces and cities of Sichuan Province (Gansu and
Chongqing) also contributed to the carbonyl compounds of the CPUA.

The potential sources of carbonyl compounds at different pollution stages at the

Chengdu Institute of Environmental Sciences site during the observation period are
shown in Fig. 9. It can be seen from the figure that there are differences in the potential
sources of carbonyl compounds among different pollution stages at the CDHKY site.
The concentration of local carbonyl compounds in CDHKY was high during the early
observation period and EP1, which existed local sources, and was also affected by the
northern airflow, and carbonyl compounds was also affected by the transport from MY,
DY and other northern regions. Under the effect of the continuous northern airflow, the
local source emissions decreased during EP1, and the potential source of carbonyl
compounds changed to from the junction between CDHKY and ZY. During EP3, under
the combined influence of the western airflow, the contribution of transport from SN
and ZY to carbonyl compounds increased, while emissions from local sources also
increased.

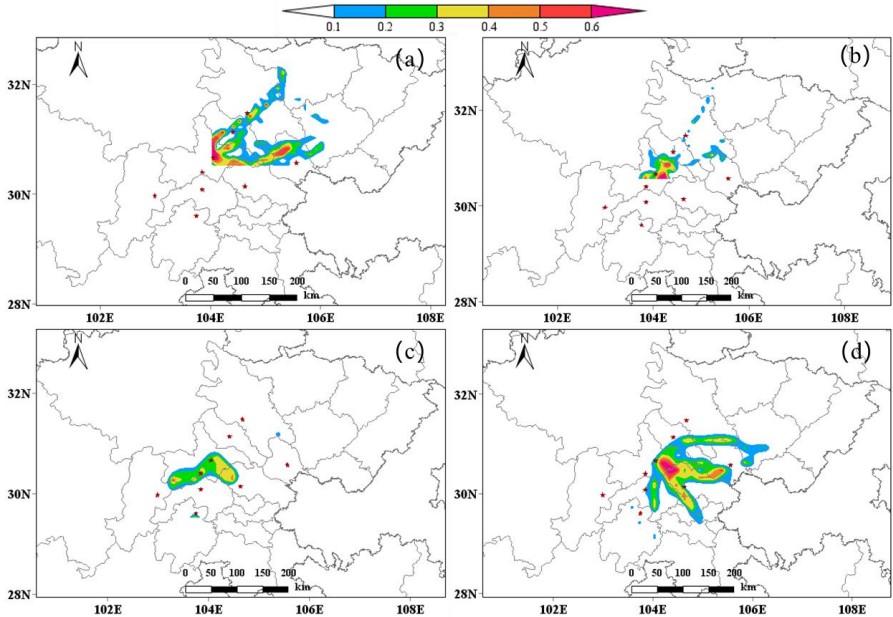


**Figure 9.** Analysis of potential sources of carbonyl compounds at different periods at the CDHKY



642 site during the observation period (a) August 4th-6th (b) August 7th-9th (c) August 10th-13th (d)

643         August 15th-18th

644 **4. Conclusions**

645    During a concurrent atmospheric observation campaign conducted at nine sites in

646 the CPUA from August 4th to 18th, 2019, three regional heavy ozone pollution episodes,

647 labeled EP1 to EP3, were observed. This study extensively examines the concentration

648 variations, atmospheric chemical reactivity, and sources of carbonyls during this period.

649 The average total concentrations of 15 carbonyl compounds across the nine sites within

650 eight cities of the CPUA were measured at $17.35 \pm 5.31$ ppb. Spatial analysis revealed

651 a positive correlation between carbonyl levels and ozone concentrations, particularly

652 concentrated around Chengdu in both northern and southern directions. Formaldehyde

653 (36.4%-64.3%), acetone (12.4%-28.1%), and acetaldehyde (8.2%-47.3%) constituted

654 the predominant species by volume concentration. Intriguingly, Chengdu exhibited FAT

655 concentrations surpassing national and international levels, indicating heightened levels

656 compared to other regions. Diurnal variations showed peaks during the day and lows at

657 night, with notable spikes on ozone pollution days. A distinctive "weekend effect" was

658 observed, particularly evident in carbonyl compounds associated with motor vehicle

659 emissions, such as acetaldehyde and acetone, peaking during morning rush hours and

660 nighttime on weekends. This suggests significant contributions from both daytime

661 photochemical processes and nighttime vehicular emissions to carbonyl compounds. At

662 the MY site, 48.6% of the total volatile organic compounds (VOCs) ozone formation

663 potential (OFP) was attributed to the 15 carbonyl compounds, emphasizing their

664 substantial impact on ozone formation, especially formaldehyde and acetaldehyde.

665    Ground-level observations of FNR were utilized to assess the sensitivity of

666 ground-level ozone formation. FNR from ground-level observations were used to

667 determine the sensitivity of ground-level ozone formation. Analysis of FNR revealed

668 that sites experiencing heavy ozone pollution exhibited lower FNRs, indicating a



VOCs-limited regime, while sites with lighter ozone pollution were categorized into a
transitional regime. Carbonyl compound sources include primary emissions and
secondary formation processes. Multivariate linear regression quantitatively analyzed
formaldehyde, acetaldehyde, and acetone sources. Secondary formation contributed
over 30% on average to formaldehyde, acetaldehyde, and acetone, despite primary
emissions being primary sources. OBM modeling revealed that formaldehyde and
acetaldehyde primarily originated from the secondary formation of alkenes and BVOCs,
while acetone mainly stemmed from the secondary formation of alkanes. Furthermore,
it is recommended to establish a scientific control mechanism for both NOx and VOCs,
with special attention to formaldehyde, acetaldehyde, and acetone, and their alkenes
precursors. Additionally, considering the regional nature of pollution, this study
suggests that carbonyl compound pollution is influenced by mutual transport among
cities within the CPUA, notably along the MY-DY-CDHKY route. Establishing a
collaborative prevention and control mechanism among cities within the CPUA and
neighboring provinces and cities is crucial to effectively address carbonyl compounds
and ozone pollution in the region in the future.

**Data availability.** Observational data including meteorological parameters and air
pollutants used in this study are available from the corresponding authors upon request
(lihong@craes.org.cn).

**Author contributions.** Hong Li and Jiemeng Bao designed this study. Xin Zhang,
Zhenhai Wu, Jiemeng Bao, Li Zhou, Qinwen Tan, and Fumo Yang coordinated the
selection of field observation sites, including locations for both VOCs and carbonyls
grid sampling. Qinwen Tan and Hefan Liu supported the collection of carbonyls at one
site. Zhenhai Wu and Xin Zhang assisted in carbonyls sampling; Xin Zhang and
Yunfeng Li assisted in carbonyls sample analysis and data collection. Li Zhou and
Hefan Liu organized the analysis of VOCs measurements. Jun Qian, Junhui Chen, and



Liqun Deng provided support in project funding application. Jiemeng Bao performed
the data analysis and wrote the paper with contributions from all co-authors; Hong Li
reviewed the paper, provided comments and finalized it.

**Competing interests.** The contact author has declared that none of the authors has any
competing interests.

**Acknowledgments.** The authors would like to express their sincere appreciation to
Keding Lu and Xin Li of Peking University for their organization of the intensive field
observation experiment on the formation mechanisms of photochemical pollution in
summer in the CPUA of China. They also want to show their deep gratitude to Yulei
Ma, Tianli Song, Xiaodong Wu, Ning Wang, and He Zijun Liu of Sichuan University,
as well as Xin Zhang (female) and Hefan Liu of Chengdu Academy of Environmental
Protection Sciences for their help in sampling. They are also grateful to Liping Liu of
Sichuan Agricultural University in Ya'an City, Kaiyao Lv of Mianyang High-tech Zone
Management Committee, Yong Xiao of Deyang Municipal Education Bureau, Ying Ni
of Meishan Ecological Environment Bureau, Aihua Zou of Leshan Ecological
Environment Bureau, and Chuhan Wang of the Chinese Academy of Environmental
Sciences for their substantial support during field observations. Special thanks to Zhen
He and Manfei Yin of the Chinese Academy of Environmental Sciences for their
assistance in analyzing samples from the XJ site.

**Financial support.** This research has been supported by the Research Project on
Analysis of Multiple Causes of Atmospheric Ozone Pollution in Urban Agglomerations
of Chengdu Plain and Development of Management, Prevention, and Control System
of Sichuan Academy of Environmental Sciences (No. 510201201905430).

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
