# Peer review of "Exploring the Crucial Role of Atmospheric Carbonyl Compounds in Regional Ozone heavy Pollution: Insights from Intensive Field Observations and Observation-based modelling in the Chengdu Plain Urban Agglomeration, China"

_EGUsphere, 2024_

## Author Response (AR1)

**Response to the Comments of the Reviewers**

**---For the manuscript "egusphere-2024-1204"**

Dear Editor and Reviewers,

We acknowledge the constructive comments and encouragement of the reviewers and are grateful for the efficient service of the editor. Here, we submit our revised manuscript titled "Exploring the Crucial Role of Atmospheric Carbonyl Compounds in Regional Ozone heavy Pollution: Insights from Intensive Field Observations and Observation-based modelling in the Chengdu Plain Urban Agglomeration, China" (Manuscript ID: egusphere-2024-1204), along with a thorough, point-by-point response to each comment raised by the reviewers. The revisions to the manuscript are highlighted in blue text in the attached "Response to the Comments of the Reviewers." Additionally, we have provided a clean version of the revised manuscript as required. We greatly appreciate the reviewers' insightful comments and valuable suggestions, which have significantly improved the quality of our manuscript.

Sincerely yours,

Authors of the manuscript egusphere-2024-1204

Corresponding author: Hong Li (lihong@craes.org.cn)

First author: Jiemeng Bao (2301112284@stu.pku.edu.cn)

Oct. 25, 2024

**Response to the Reviewer #1**

**Comment 1:** Poor use of the literature: There is very poor coverage of the early work on carbonyls, which results in the appearance that carbonyls have only just been discovered in the literature. Guenther et al. 2012 is not an appropriate reference for the photolysis of carbonyls (even a textbook such as Seinfeld and Pandis would be better). The introduction should use more modern references to point out how understanding has progressed since the early studies, not to stand-in for the earlier work.

**Response:**

We have expanded the introduction to include a more comprehensive review of early work on carbonyl compounds. The reference to Guenther et al. (2012) for photolysis has been replaced with more appropriate sources, such as Seinfeld and Pandis (2016), to better represent the progression in the understanding of carbonyl chemistry. The revised introduction now reflects both early foundational studies and more recent advances, improving the context of carbonyl compounds' roles in ozone formation.

Lines 47-94:

" Atmospheric carbonyl compounds play a pivotal role in tropospheric chemistry, acting as crucial precursors to both ozone ($O_3$) and secondary organic aerosols (SOA), a fact recognized for decades (Altshuller, 1993; Grosjean and Seinfeld, 1989). Their importance has been confirmed by numerous studies over the years(Guo et al., 2004; Hallquist et al., 2009; Wang et al., 2020; Ye et al., 2021; Coggon et al., 2019), highlighting their significant contribution to atmospheric photochemistry and air pollution. Over the past two decades, severe air pollution in China has driven substantial research efforts to understand the contributions of carbonyl compounds to these environmental challenges. Studies have shown that photolysis of carbonyl compounds is a major source of ROX radicals (Grosjean and Seinfeld, 1989; Zhang et al., 2016). These compounds can be photolyzed and react with OH radicals to form a large number of HO2 and RO2 radicals, which increase

the atmospheric oxidation capacity and participate in the NOx photochemical cycle, leading to ozone formation (Zhang et al., 2016; Meng et al., 2017). Additionally, dialdehydes such as glyoxal and methylglyoxal undergo heterogeneous reactions with aqueous particulate matter, rapidly forming SOA (Lou et al., 2010; Xue et al., 2016; Yuan et al., 2012). Ambient carbonyl compounds not only affect the environment but also pose direct health risks to humans. They can harm ecosystems through deposition and adsorption processes (Yang et al., 2018). They also pose direct health risks to humans, including sensitization, carcinogenesis, and mutagenicity (Fuchs et al., 2017).

Recent research has increasingly focused on understanding the spatial and temporal variability of carbonyl compounds in highly polluted regions, particularly in China, where rapid industrialization has led to severe air quality challenges. Xue et al. (2013) and Duan et al. (2012) reported typical ambient concentrations of carbonyl compounds ranging from a few $\mu g \cdot m^{-3}$ to tens of $\mu g \cdot m^{-3}$ in urban areas, depending on the specific compounds and regions studied. For example, formaldehyde concentrations in highly polluted areas can exceed 10 $\mu g \cdot m^{-3}$. Shen et al. (2013) and Fu et al. (2008) observed significant diurnal variation, with higher concentrations of carbonyl compounds during the daytime, particularly in the afternoon, driven by photochemical production. Concentrations can increase by as much as 50-100% during peak sunlight hours compared to nighttime levels. Pang and Mu (2006) and Rao et al. (2016) identified key sources of carbonyl compounds, including vehicular emissions, industrial activities, and secondary formation from VOC oxidation in the atmosphere. In urban environments, vehicular emissions are often a dominant primary source, while secondary formation contributes significantly during daytime due to photochemical processes. The results highlight severe and spatiotemporal variations of carbonyl pollution in China. High levels are found mainly in the North China Plain(NCP), the Yangtze River Delta(YRD), and the Pearl River Delta(PRD)(Duan et al., 2008; Shao et al., 2009; Tan et al., 2018; Wang et al., 2018; Xue et al., 2014, 2013; Yang et al., 2017). Urban areas generally exhibit higher carbonyl levels than suburban and rural areas due to human activities(Xue et al., 2013). Despite the progress made, significant gaps remain in understanding the spatiotemporal distribution and source

apportionment of carbonyl compounds, particularly in urban agglomerations. Existing research has primarily focused on urban areas in rapidly developing regions like the NCP, YRD, and PRD. Moreover, studies have often emphasized the overall role of VOCs in ozone pollution, with less attention given to specific carbonyl compounds and their individual contributions to atmospheric oxidation capacity and ozone formation (Meng et al., 2017)."

**Comment 2:** Poor framing of research question (lines 86-88). The research gap identified here is too broad to be very meaningful: the literature is full of evaluations of the specific roles of carbonyls in ozone production. Again, in lines 89-105, especially lines 103-105, the precise research gap is not identified with sufficient precision.

**Response:** We have refined the framing of the research question to focus on whether it is the **abundance** of carbonyls or specific **additional chemistry** that explains their importance in ozone formation in this context. This adjustment provides a clearer and more focused research gap that addresses whether current knowledge can fully account for the observed ozone production in the Chengdu Plain.

Lines 95-124:

"Monitoring carbonyl compounds in the atmosphere is challenging due to their typically low concentrations (ppt-ppb levels), necessitating highly sensitive analytical methods. The diversity of carbonyl compounds, including multiple isomers, requires highly selective analytical techniques for differentiation. Current measurement technologies limit our understanding of the spatiotemporal distribution of carbonyl compounds, affecting the accurate assessment of their environmental behavior, sources, and transport (Xue et al., 2013; Sahu and Saxena, 2015). While numerous studies have explored the role of carbonyl compounds in ozone production, many focus on general mechanisms rather than specific compounds or regional variations (Atkinson and Arey, 2003; Monks et al., 2015).

Atmospheric carbonyl compounds originate from both primary and secondary sources (Pang and Mu, 2006; Rao et al., 2016). Primary sources include the

incomplete combustion of fossil fuels and biomass, industrial emissions, emissions from the catering industry, and releases from plants. Secondary sources arise from the atmospheric photochemical oxidation of VOCs (Xue et al., 2013), particularly alkenes, aromatics, and isoprene, which typically dominate the secondary formation of carbonyls. However, distinguishing between primary and secondary contributions remains challenging. Existing source apportionment methods, such as characteristic species ratios and multiple linear regression, often lack the resolution to differentiate these sources accurately, especially for non-vehicular emissions and secondary formation. The limitations of these methods underscore the need for more advanced approaches to better quantify the secondary formation mechanisms of carbonyl compounds and their regional impact on ozone formation. Despite significant advancements in studying atmospheric carbonyls, key gaps remain in understanding their precise spatiotemporal distribution and source apportionment. Specifically, there is a need for studies that examine how carbonyls vary across different environments— urban, suburban, and rural—and during varying pollution events. Without such targeted analysis, our understanding of the behavior of carbonyl compounds and their contribution to ozone pollution remains incomplete, particularly in regions experiencing severe pollution.*"*

**Comment 3:** When the research has been properly framed (addressing points 1 and 2, above), the Results & Discussion and Conclusions sections should be modified accordingly.

**Response:** Based on the revised research question, we have made corresponding changes to the Results & Discussion and Conclusions sections. These sections now emphasize whether the importance of carbonyls is due to their abundance or other specific chemical mechanisms, as observed in the study region.

**Comment 4:** L39, Abstract: I think "alkenes and alkanes being important secondary precursors of carbonyls" should be "alkenes and alkanes being important precursors of secondary carbonyls" – because it is the carbonyls that are secondary, not the alkanes and alkenes.

**Response:** We have revised the sentence to "alkenes and alkanes being important precursors of secondary carbonyls" to clarify that the carbonyls are secondary.

**Comment 5:** Ll48-49. It is not sufficient to support the introductory statement with a citation from 2004. The importance of carbonyls to ozone and SOA has been known for decades and described by earlier authors.

**Response:** The introductory statement has been updated with a more recent reference that reflects the ongoing importance of carbonyl compounds in ozone and secondary organic aerosol (SOA) formation.

Lines 49-53:

"Their importance has been confirmed by numerous studies over the years(Guo et al., 2004; Hallquist et al., 2009; Wang et al., 2020; Ye et al., 2021; Coggon et al., 2019), highlighting their significant contribution to atmospheric photochemistry and air pollution."

**Comment 6:** Ll63-65. Please give some indication of the concentrations and the size of the diurnal variation reported in these papers so that the reader can immediately compare with what is in the current paper.

**Response:** We have added quantitative data on the concentrations and diurnal variations reported in the cited papers, allowing readers to directly compare the values with those from our study.

Lines 82-88:

"Recent research has increasingly focused on understanding the spatial and temporal variability of carbonyl compounds in highly polluted regions, particularly in China, where rapid industrialization has led to severe air quality challenges. Xue et al. (2013) and Duan et al. (2012) reported typical ambient concentrations of carbonyl compounds ranging from a few $\mu g \cdot m^{-3}$ to tens of $\mu g \cdot m^{-3}$ in urban areas, depending on the specific compounds and regions studied. For example, formaldehyde concentrations in highly polluted areas can exceed 10 $\mu g \cdot m^{-3}$."

**Comment 7:** Ll80-81: This statement is a bit too strong; it is perfectly possible to

measure formaldehyde (which the abstract says is ~50% of the carbonyls of concern in this study) from space.

**Response:** We have revised the statement to "Urban areas generally exhibit higher carbonyl levels than suburban and rural areas due to human activities."

**Comment 8:** Ll94-97: a statement as strong as this requires support from the literature. Similarly, the sentence following on lines 98-99.

**Response:** We have provided additional references to support the strong statements made in these sections.
Lines 91-102:

"Moreover, studies have often emphasized the overall role of VOCs in ozone pollution, with less attention given to specific carbonyl compounds and their individual contributions to atmospheric oxidation capacity and ozone formation (Meng et al., 2017).

Monitoring carbonyl compounds in the atmosphere is challenging due to their typically low concentrations (ppt-ppb levels), necessitating highly sensitive analytical methods. The diversity of carbonyl compounds, including multiple isomers, requires highly selective analytical techniques for differentiation. Current measurement technologies limit our understanding of the spatiotemporal distribution of carbonyl compounds, affecting the accurate assessment of their environmental behavior, sources, and transport (Xue et al., 2013; Sahu and Saxena, 2015)."

**Comment 9:** L135: OBM should be defined on first use.

**Response:** The term "OBM" (Observation-Based Model) is now defined at its first appearance in the manuscript.

**Comment 10:** Figure 1 caption is insufficiently detailed and should at least say what is shown on left and right-hand panels. It is not clear what "9 mg.m$^3$/grid" means, especially since the colour on the map appear to be interpolated to a smooth surface rather than gridded.

**Response:** The caption for Figure 1 has been expanded to clarify the content of the left and right images. We have also explained the legend in the right panel and ensured the description matches the presentation in the figure. "9 μg·m⁻³/grid" refers to each color representing 9 μg·m⁻³.

Lines 183-192:

"Figure 1. Distribution of sampling sites. The left panel shows the elevation map of the Sichuan Basin, highlighting the geographical features of the region, with elevation data sourced from the Geospatial Data Cloud (https://www.gscloud.cn/#page1/2). The right panel presents the spatial distribution of ozone concentrations in the CPUA during the observation period (August 4–18, 2019), with ozone data obtained from national control stations near each sampling site. Black dots represent the locations of the sampling sites, labeled as follows: MY (Mianyang), DY (Deyang), CDHKY (Chengdu Environmental Science Research Institute), XJ (Xinjin), SN (Suining), ZY (Ziyang), MS (Meishan), YA (Ya'an), and LS (Leshan). The color bar in the top left corner corresponds to interpolated ozone concentrations, with each color representing a concentration gradient."

**Comment 12:** L201: is "TO-15" a method or a chemical or family of chemicals? Please clarify the two uses in this paragraph and define PAMS on line 211.

**Response:** We have clarified that TO-15 refers to a method for measuring volatile organic compounds (VOCs), and we have defined PAMS (Photochemical Assessment Monitoring Stations).

Lines 241-252:

"The atmospheric VOCs were analyzed using the TO-14 and TO-15 methods, which are recommended by the US EPA. These methods involve frozen preconcentration coupled with gas chromatography and mass spectrometry (GC-MS). TO-15 is a method for detecting and quantifying a wide range of VOCs from air samples. The VOCs were pre-concentrated by the Entech7100 system at a low temperature, then quantified by an Agilent GC-MS. During the sample analysis, four internal standard gases (bromochloromethane, 1,4-difluorobenzene, chlorobenzene-d5,

and 4-bromofluorobenzene) were used. A multi-point calibration curve was created using a standard gas containing 118 VOCs, including PAMS compounds, TO-15 target analytes, and carbonyl compounds. PAMS (Photochemical Assessment Monitoring Stations) compounds are a subset of hydrocarbons known to contribute to ozone formation, such as ethane, ethylene, propane, and others."

**Comment 13:** L203: "mass chromatography" should be "mass spectrometry"

**Response:** "Mass chromatography" has been corrected to "mass spectrometry."

**Comment 14:** L227 "Inferring ozone formation sensitivity" is better English

**Response:** We have rephrased the sentence to "Ozone formation sensitivity" to improve clarity.

**Comment 15:** Ll243-250. Better to use lower case k for rate constants so as not to confuse with equilibrium constants. The rate is not given by eq (1) but by the right-hand side of eq (1) times the concentration of OH.

**Response:** We have changed the rate constant notation to lowercase 'k' and clarified that the rate is given by the right-hand side of eq (1) multiplied by the concentration of OH.

**Comment 16:** Ll251-255: What are the units of OFP and MIR_i?

**Response:** The units for OFP (Ozone Formation Potential) and MIR (Maximum Incremental Reactivity) have been added for clarity.

**Comment 17:** L326: Is there a citation, url, or business address for the MeteoInfo software and Trajstat plug-in?

**Response:** We have included a URL for the MeteoInfo software and Trajstat plug-in in the text.
Lines 383-385:

"The effects of long-distance air mass transport on the pollution of carbonyl compounds in the CPUA were studied using MeteoInfo software and TrajStat plug-in

(http://www.meteothink.org/downloads/index.html ).”

**Comment 18:** Ll345-346. Ozone is insoluble, so a little more explanation of why precipitation alleviated ozone pollution is needed here.

**Response:** We have provided a more detailed explanation of how precipitation alleviates ozone pollution, despite ozone's insolubility, by discussing the removal of ozone precursors from the atmosphere through wet deposition. Although ozone itself is not easily removed by rain, precipitation reduces ozone pollution by washing away its precursors, such as nitrogen oxides (NOx) and volatile organic compounds (VOCs), decreasing sunlight exposure, and enhancing atmospheric dispersion.

**Comment 19:** Fig 3. I will not insist, but if you have the opportunity to re-draw this figure using colour scales that are easier for those with colour-blindness, that would be good.

**Response:** We have adjusted the color scale in Figure 3 to make it more accessible for color-blind readers.

**Comment 20:** L385. I think this should be "the average total concentration of the 15 carbonyls…"

**Response:** We have corrected the sentence to read "the average total concentration of the 15 carbonyls..”

**Comment 21:** Table 1. The caption should state where the reader can find an explanation of the column headings. The caption should read "Daily mean +/- standard error…" (or standard deviation, whichever it is).

**Response:** The caption for Table 1 has been revised to state: "Daily mean $\pm$ standard error (or standard deviation, as appropriate)...” and now includes an explanation of the column headings.

**Comment 22:** L397ff: please do not switch from names of cities to acronyms inconsistently. It is best to remind the reader by using both name and acronym at first, before using just one.

**Response:** We ensured consistency in city names and abbreviations throughout the manuscript and clarified the relationship between site names and cities when first mentioned in Figure 1.

**Comment 23:** L423: specie is not the singular of species – replace with 'carbonyl'

**Response:** We have replaced "specie" with "species" to correct this grammatical error.

**Comment 24:** L428: replace 'concentrations' with 'measured', since not all carbonyls have been measured.

**Response:** We have revised the sentence to "measured" to clarify that not all carbonyls have been measured.

**Comment 25:** L431: funny that MACR appears but not methyl vinyl ketone, since both are produced in roughly equal measure from isoprene. Acetone is the only ketone that appears in Table 1. Is that to be expected?

**Response:** MVK and other ketones were not detected, possibly due to the limitations of the TO-15 method. The TO-15 method is primarily used to detect volatile organic compounds (VOCs) in the air, with a focus on lower molecular weight VOCs such as hydrocarbons, halogenated hydrocarbons, and aromatics. Although the TO-15 method can detect certain aldehydes and ketones, its detection sensitivity and efficiency may not be sufficient for specific compounds like MVK and other ketones.

**Comment 26:** L466: deposition would also play a part in a diurnal cycle of this kind.

**Response:** We have mentioned deposition as a contributing factor to the diurnal cycle of carbonyls, in addition to their chemical production and loss processes.

**Comment 27:** Figure 5. It is difficult to interpret this figure without a better caption. Unit for L_OH should be $s^{-1}$ (i.e., lowercase s). Negative indices and solidus ('/')

notation should not be used together. Negative indices should be used consistently throughout the document.

**Response:** We have revised the figure caption and unit labels to ensure consistency in the use of notation and correct units ($s^{-1}$).

**Comment 28:** L524: every compound listed in Table 1 is an aldehyde except acetone, so this sentence presumably means simply that acetone is at higher concentration?

**Response:** We have clarified that the term "aldehyde" refers to acetaldehyde in this context.

**Comment 29:** Figure 9: the caption does not explain the figure sufficiently well. The maps could be 'zoomed' closer into the area of interest.

**Response:** Thank you for your feedback. We have revised the figure caption to provide a clearer explanation. As for zooming further into the area of interest, we prefer to maintain the current scale, as it encompasses all eight cities in the Chengdu Plain Urban Agglomeration. Further zooming might result in an incomplete map of the region.

**Response to the Reviewer #2**

**Comment 1:** Since many of the values reported in the paper are averages across multiple sites, it is recommended to include the standard deviations. For example, Lines 27-28, Lines 340-341, Lines 425-426, and other relevant sections.

**Response:**

We thank the reviewer for the insightful suggestion regarding the inclusion of standard deviations for values reported as averages across multiple sites. We have revised the manuscript accordingly to enhance the clarity and robustness of our findings.

Lines 27-28:

"Throughout the study, the total mixing ratios of 15 carbonyls ranged from $10.70\pm4.16$ to $35.18\pm13.37$ ppbv, in which formaldehyde (48.1%), acetone (19.9%), and acetaldehyde (17.5%) were most abundant within the CPUA."

**Comment 2:** "The ozone formation sensitivity for sites experiencing severe ozone pollution were classified as VOCs-limited regime, while others were categorized as transitional regime". This statement is ambiguous. Does this refer to sites with varying degrees of ozone pollution, or does it pertain to the same site experiencing different stages of an ozone episode?

**Response:**

Thank you for your valuable feedback. To clarify, the statement refers to the same site experiencing different stages of an ozone episode. We have revised the sentence to: "Sites with higher average ozone concentrations during observations were mainly in the VOCs-limited regime, while others were in the transitional regime." This change enhances clarity and accurately reflects the intended meaning.

**Comment 3:** Lines 177-178: Why were 6 samples collected over three days? Was it because these were ozone pollution days? Does the inconsistency in the VOCs and carbonyls collection times cause uncertainties in subsequent analysis?

**Response:**

Thank you for your comments. The increased sampling frequency for VOCs on August 11, 12, and 16 was implemented specifically because all eight cities in the Chengdu Plain Urban Agglomeration (CPUA) experienced ozone pollution (from mild to severe levels) on these dates. Collecting six samples per day during these pollution events allowed for better capturing of ozone-related VOC variations throughout the day.

In the subsequent analysis, the VOCs data from these three days was used in an observation-based model (OBM) to calculate the relative incremental reactivity (RIR) for formaldehyde, acetaldehyde, and acetone. Using the OBM classification approach, we evaluated the contributions of anthropogenic VOCs (such as alkanes, alkenes, alkynes, and aromatics) and biogenic VOCs (such as isoprene) to the formation of these carbonyl compounds. This targeted approach during periods of elevated ozone levels ensures that our findings on VOCs' contributions to formaldehyde, acetaldehyde, and acetone formation are representative of typical ozone pollution conditions in CPUA.

We acknowledge that the difference in sampling schedules between VOCs (collected twice daily or six times during pollution episodes) and carbonyl compounds (collected every two hours) could introduce temporal variation. However, because the focus of our analysis was on capturing representative VOCs contributions to formaldehyde, acetaldehyde, and acetone formation during peak ozone pollution periods, the approach remains robust. Additionally, the alignment of VOC and carbonyl data during identified pollution episodes on August 11, 12, and 16 provides confidence in the reliability of the RIR (Relative Incremental Reactivity) results under typical pollution conditions in the CPUA.

Lines 475-480:

"From August 4th to 18th, 2019, two VOCs samples were collected each day at each site, at 8:00-9:00 and 14:00-15:00 (no samples were taken under special weather conditions, such as rain). On August 11th, 12th and 16th , six samples were collected

per day to capture diurnal variations under ozone pollution events, at the following times: 8:00-9:00, 10:00-11:00, 12:00-13:00, 14:00-15:00, 16:00-17:00, and 18:00-19:00."

**Comment 4:** Some subtitles are not appropriate. For example, "Ambient levels comparison" is suggested to revise to "Ozone pollution assessment criteria", and it is recommended to delete Lines 221-226. "Ozone formation sensitivity inferring" should be changed to "Ozone formation sensitivity", "Secondary formation mechanism investigation" is not suitable as a subtitle.

**Response:**

Thank you for your insightful suggestions regarding the subtitles. We have made the following revisions:

The subtitle "Ambient levels comparison" has been changed to "Ozone pollution assessment criteria." We have changed "Ozone formation sensitivity inferring" to "Ozone formation sensitivity." The subtitle "Secondary formation mechanism investigation" has been revised to "Exploration of Secondary Formation Mechanisms." Additionally, the title "Investigation of secondary formation mechanism of key carbonyl compounds" has been updated to "Exploration of secondary formation mechanism of key carbonyl compounds."

**Comment 5:** Please clarify the mechanism used by the OBM model.

**Response:** Thank you for your constructive feedback. We have added a detailed description of the mechanism used by the OBM model in the Methods section of the manuscript.

Lines 299-312:

"The Observation-Based Model (OBM) is a box model that uses actual observational data to evaluate the sensitivity of secondary pollutant formation mechanisms to their precursor emissions. By constraining the model with atmospheric

observation data, typical secondary pollutants and parameters such as NOX, SO2, CO, VOCs, temperature, humidity, pressure, and JNO2 are input into the model as hourly observational data to calculate the chemical formation and consumption of secondary pollutants and free radicals. In this study, the OBM model used the Master Chemical Mechanism (MCM) (v3.3.1, mcm.leeds.ac.uk), which is a nearly detailed chemical mechanism that describes the chemical processes of 143 VOC species from emission to degradation in the atmosphere, including approximately 6,700 species and 17,000 inorganic and organic reactions. The MCM chemical mechanism can simulate atmospheric photochemical reaction processes under near-real conditions and calculate the concentrations of highly reactive species, quantifying the reaction rates of all species involved."

**Comment 6:** References are needed in the RIR calculation.

**Response:** Thank you for pointing this out. We have added the relevant references to support the calculation of Relative Incremental Reactivity (RIR) in the manuscript. Lines 299-312:

"Relative Incremental Reactivity (RIR) was first used by Cardelino and Chameides (1995) to simulate the response of ozone to precursor changes through scenario tests using box model calculations."

**Comment 7:** The measurement instruments for NO2 and CO (as shown in Figure 2 and Figure 3), as well as the time resolutions, were not introduced in the method section.

**Response:** Thank you for your valuable feedback. We have added a description of the measurement instruments for $NO_2$ and CO, along with their respective time resolutions, to the Methods section. Additionally, we have included information about the source of the meteorological data. These enhancements improve the clarity and completeness of our methodology.

Lines 170-181:

"Ozone concentrations were measured using the UV absorption method with a Thermo $O_3$ analyzer (Model 49i), with data sourced from national control stations near each sampling site. Nitrogen dioxide ($NO_2$) was measured by chemiluminescence following chemical conversion to nitric oxide (NO) using a molybdenum catalyst; however, this method is known to have interferences from other NOz species. Carbon monoxide (CO) was measured via infrared absorption with a Thermo instrument (Model 20). All Thermo instruments were carefully maintained and calibrated daily at 01:00 to ensure measurement accuracy. Measurements for ozone, $NO_2$, and CO were collected with a time resolution of one hour. Simultaneously, meteorological parameters—temperature, relative humidity (RH), wind speed, and direction—were recorded at each observation site using an automatic weather station (PC-4, JZYG, China), also at a one-hour resolution."

**Comment 8:** Line 357: Does "O3-8" refer to the "maximum daily average 8h ozone concentration"?

**Response:** Yes, "$O_3$-8" refers to the "maximum daily average 8-hour ozone concentration." Thank you for your inquiry, and we appreciate your attention to detail. Line 418:

"After observing the spatial distribution of ozone concentration during EP1, it's evident that the severity of pollution reached heavily polluted levels, with Chengdu recording an MDA8 concentration of 297 $\mu g \cdot m^{-3}$ on August 7th."

**Comment 9:** Lines 449-451: Drawing a conclusion based solely on a comparison with 2010 seems insufficient.

**Response:** We appreciate your valuable feedback regarding the conclusion drawn from the comparison with 2010 data. We have revised this section to provide a more objective description of the high concentrations of carbon-based compounds in Chengdu, thus strengthening our analysis and conclusions.

Lines 512-516:

"The consistently high levels of carbonyl compounds observed in Chengdu, both in 2010 and our current study, indicate that the city likely experiences higher concentrations of these pollutants compared to other regions across the country. However, more extensive temporal data would be beneficial to fully validate this pattern at a national scale."

**Comment 10:** The diurnal variation of the carbonyl compounds on weekdays and weekends appears to be irrelevant. I suggest removing this paragraph.

**Response:** Thank you for your suggestion. We have removed the paragraph discussing the diurnal variation of carbonyl compounds on weekdays and weekends as recommended.

**Comment 11:** Line 519: How is the positive correlation observed?

**Response:** We acknowledge that the statement "LOH and OFP during different pollution periods show a strong positive correlation with the severity of ozone pollution" may lead to misunderstanding, as the existing data do not provide sufficient evidence to directly support this positive correlation. We have removed this statement from the manuscript and adjusted the relevant discussion to avoid overinterpretation of the results and to maintain a rigorous presentation of our findings.

**Comment 12:** Line 540: Which figure does "Fig.3.9"refer to?

**Response:** Thank you for your observation. This reference was a mistake; it should refer to Figure 6. We have corrected this in the manuscript.

Lines 579-580:

"The change of $O_3$ formation sensitivity of each site in the CPUA during the observation period is shown in Fig.6."

**Comment 13:** Fig. 1: Please include the data sources for both figures.

**Response:** Thank you for your suggestion. We have added the data source

information to the figure captions for both figures in the manuscript.

Lines 183-192:

"Figure 1. Distribution of sampling sites. The left panel shows the elevation map of the Sichuan Basin, highlighting the geographical features of the region, with elevation data sourced from the Geospatial Data Cloud (https://www.gscloud.cn/#page1/2). The right panel presents the spatial distribution of ozone concentrations in the CPUA during the observation period (August 4–18, 2019), with ozone data obtained from national control stations near each sampling site. Black dots represent the locations of the sampling sites, labeled as follows: MY (Mianyang), DY (Deyang), CDHKY (Chengdu Environmental Science Research Institute), XJ (Xinjin), SN (Suining), ZY (Ziyang), MS (Meishan), YA (Ya'an), and LS (Leshan). The color bar in the top left corner corresponds to interpolated ozone concentrations, with each color representing a concentration gradient."

**Comment 14:** For Figure 3a, how is the average O3 concentration calculated? Is it the average over the entire day, or is it the maximum daily average 8h ozone (MDA8) average? Additionally, O3 and NO2 seem relatively consistent in this figure. How about the temporal and spatial variations of VOCs and carbonyl compounds?

**Response:** The average $O_3$ concentration presented in Figure 3a refers to the average of the maximum daily average 8-hour ozone (MDA8) values. Regarding the temporal and spatial variations of volatile organic compounds (VOCs) and carbonyl compounds, we have illustrated this information in Figure 2.

**Comment 15:** Figure 6: Please use different markers to distinguish EP1, EP2 and EP3.

**Response:** In Figure 6, we have used different colors to represent each monitoring site. Additionally, we employed distinct markers for each site during the EP1 to EP3 phases, connected by dashed lines with arrows indicating the progression from EP1 to EP3.

---

## Author Response (AR2)

**Response to the Comments of the Reviewers**

**---For the manuscript "egusphere-2024-1204"**

Dear Editor and Reviewers,

We acknowledge the constructive comments and encouragement of the reviewers and are grateful for the efficient service of the editor. Here, we submit our revised manuscript titled "Exploring the Crucial Role of Atmospheric Carbonyl Compounds in Regional Ozone heavy Pollution: Insights from Intensive Field Observations and Observation-based modelling in the Chengdu Plain Urban Agglomeration, China" (Manuscript ID: egusphere-2024-1204), along with a thorough, point-by-point response to each comment raised by the reviewers. The revisions to the manuscript are highlighted in blue text in the attached "Response to the Comments of the Reviewers." Additionally, we have provided a clean version of the revised manuscript as required. We greatly appreciate the reviewers' insightful comments and valuable suggestions, which have significantly improved the quality of our manuscript.

Sincerely yours,

Authors of the manuscript egusphere-2024-1204

Corresponding author: Hong Li (lihong@craes.org.cn)

First author: Jiemeng Bao (2301112284@stu.pku.edu.cn)

Nov. 15, 2024

**Response to the Reviewer #1**

**Comment 1:** Line 22: change to reflect more precise research questions: i.e., "whether it is the abundance of carbonyls or specific additional chemistry that explains their importance in ozone formation in this context."

**Response:**

Thank you for your valuable comment. In response, we have revised this sentence to reflect a more precise research question. We now explicitly examine whether it is the abundance of carbonyls or the specific chemical processes involving carbonyls that explain their importance in ozone formation. This revised focus better highlights the core objective of our study, which is to assess the roles of carbonyls in the context of regional ozone pollution.

Lines 22-27:

"To determine whether the impact of carbonyl compounds on regional ozone pollution is driven by their abundance or by specific secondary chemical processes, simultaneous field observations and observation-based modelling of ambient carbonyls were conducted at nine sites within the Chengdu Plain Urban Agglomeration (CPUA), China during August 4-18, 2019, when three episodes of regional heavy ozone pollution occurred across eight cities within CPUA."

**Comment 2:** Line 27: I don't think you need the second decimal place in the averages and standard deviations reported. Reporting an average of "10.70" when the standard deviation is 4.2 doesn't add anything.

**Response:** Thank you for your helpful suggestion. We have revised the manuscript to remove the second decimal place in the averages and standard deviations where the precision of the data does not support such fine detail. This change has been applied consistently throughout the manuscript.

Lines 26-27:

"Throughout the study, the total mixing ratios of 15 carbonyls ranged from 10.7 ± 4.2 to 35.2 ± 13.4 ppbv."

**Comment 3:** Line 39. Answer your more precise research question: is it abundance or additional chemistry that matters?

**Response:** Thank you for your insightful suggestion. Based on your feedback, we have amended the conclusion to explicitly address the research question. We now emphasize that both the abundance of carbonyls and their specific chemical reactions, such as secondary formation processes, contribute significantly to ozone formation. This revision clarifies that carbonyls play a dual role in ozone pollution, driven not only by their concentrations but also by their chemical reactivity. We believe this update more effectively answers the research question and strengthens the contribution of our study to regional ozone pollution control.

Lines 28-41:

"The spatial distribution reveal that regions with higher concentrations of carbonyl compounds, such as around Chengdu, are also areas with more severe ozone pollution. Both the abundance and the chemical reactivity of carbonyl compounds, especially formaldehyde and acetaldehyde, play crucial roles in ozone formation in the CPUA. On ozone pollution days, carbonyl concentrations significantly increased by 22.8% to 66.2%.While the abundance of carbonyls is an important factor, their significant role in heavy ozone pollution within the CPUA is primarily driven by secondary chemical processes, particularly those involving alkenes and BVOCs. Sites with higher average ozone concentrations during observations were mainly in the VOCs-limited regime, while others were in the transitional regime. Additionally, the mutual transport of carbonyl compounds between cities in the CPUA suggests that regional collaboration is essential to address ozone pollution effectively. These findings offer valuable insights for developing effective strategies to control regional ozone pollution."

**Comment 4:** Line 140. change to reflect more precise research question: i.e.,

"whether it is the abundance of carbonyls or specific additional chemistry that explains their importance in ozone formation in this context."

**Response:** We appreciate your insightful comments. In response to the suggestion, we have refined the research question to better reflect the key scientific inquiry, which is whether the importance of carbonyl compounds in ozone formation is primarily driven by their abundance or whether additional specific chemical reactions involving these species contribute to their role in ozone production. This modification aligns with the study's objectives to not only quantify the presence of carbonyls but also explore their chemical behavior and reactivity in the context of ozone formation in the CPUA. We believe this refinement provides a more precise focus for the research and better addresses the complexities of photochemical pollution in the region.

Lines 139-145:

"There is still limited understanding of whether the significant roles of carbonyl compounds in ozone formation are primarily due to their abundance or whether specific chemical reactions involving carbonyls drive this process. This study aims to address these gaps by investigating the spatial distribution, sources, and specific chemical pathways of carbonyl compounds across the entire CPUA and assessing their contributions to regional ozone pollution and inter-city air transport mechanisms."

**Comment 5:** Line 194. Somewhere in this paragraph, or perhaps from line 241, it should be made clear that ketones were not well sampled, so that data for MVK, for instance, is missing.

**Response:** Thank you for the helpful suggestion. We have clarified in the revised manuscript that ketones, including methyl vinyl ketone (MVK), were not well sampled during the field observations. Consequently, data for MVK and other ketones are missing from the analysis. We have added this information to ensure the methodology is accurately represented.

Lines 240-245:

"It should be noted that while the sampling and analysis method was effective

for most carbonyl compounds, ketones, including methyl vinyl ketone (MVK), were not well sampled during the field observations. As a result, data for MVK and other ketones were missing. During the observation period, DNPH cartridges and HPLC analysis technique were used to detect a total of 15 carbonyl compounds (Table S2).*"*

**Comment 6:** Line 252. Delete end quotation mark.

**Response:** Thank you for pointing that out. We have removed the end quotation mark as suggested in the revised manuscript.

**Comment 7:** Line 307. Please make clear how many carbonyl compounds are in this version of the MCM and how many of those were measured in the study. What values were given to any VOCs, especially carbonyls, not measured?

**Response:** Thank you for your helpful comment. We have clarified the number of carbonyl compounds included in MCM v3.3.1 and indicated how many of these were measured in our study. Additionally, we have specified that for VOCs, especially carbonyls, which were not measured in our study, the MCM uses estimated values based on emission inventories, literature data, and assumptions from similar species. This ensures the robustness of our model simulations even for unmeasured species.

Lines314-318:

"In this version of the MCM, a total of 19 carbonyl compounds are included, comprising 9 aldehydes and 10 ketones. Of these, 9 carbonyl compounds were measured in this study, including formaldehyde, acetaldehyde, acetone, propionaldehyde, crotonaldehyde, butyraldehyde, isovaleraldehyde, valeraldehyde, and hexaldehyde."

Lines321-324:

"For VOCs, especially carbonyls, that were not directly measured, the MCM uses estimated values derived from emission inventories, literature data, and assumptions based on similar species to provide estimates for their concentrations and reaction rates."

**Comment 8:** Lines 321, 385, 485. Delete space before punctuation mark.

**Response:** Thank you for your careful review. We have deleted the extra spaces before the punctuation marks in lines 321, 385, and 485, as requested.

**Comment 9:** Line 388. Please specify if height is above ground level or mean sea level.

**Response:** Thank you for your suggestion. We have revised the manuscript to specify that the trajectory simulation height is 500 m above ground level (AGL), as per your request.

Lines 392-393:

  "A trajectory simulation height of 500 m above ground level (AGL) was selected."

**Comment 10:** Line 401. Since most VOCs are hydrophobic, I suggest you say "polar volatile organic compounds"

**Response:** Thank you for your helpful suggestion. We have revised the manuscript to more clearly specify examples of polar volatile organic compounds (VOCs), such as alcohols, aldehydes, ketones, and others, to provide more detail on the types of VOCs involved in ozone formation.

Lines 404-408:

  "Although ozone itself is not easily removed by rain, precipitation reduces ozone pollution by washing away its precursors, such as nitrogen oxides (NOx) and polar volatile organic compounds (VOCs), such as aldehydes, ketones, and others, decreasing sunlight exposure, and enhancing atmospheric dispersion."

**Comment 12:** Line 660. The caption ends unexpectedly. Please define the following acronyms in the caption: ALK, ALE, ACE, ARO, and BVOCs.

**Response:** Thank you for your suggestion. We have revised the caption to define the acronyms ALK, ALE, ACE, ARO, and BVOCs for clarity, ensuring that readers can easily understand the abbreviations and their corresponding meanings.

Lines 664-666:

"Figure 8. Mean RIRs of formaldehyde, acetaldehyde and acetone to different anthropogenic source VOCs (alkanes (ALK), alkenes (ALE), alkynes(ACE), aromatics (ARO))and biogenic VOCs (BVOCs)at MY, SN, ZY, YA and LS sites on August 11[th], 12[th] and 16[th]."

**Comment 13:** Line 693 or close by. Come back to research gap and address what has been learnt explicitly in terms of that research gap.

**Response:** Thank you for your helpful suggestion. In response to your comment, we have clarified how this study addresses the research gap regarding the roles of carbonyl compounds in ozone formation. We explicitly discuss how both the abundance and the specific chemical reactions involving carbonyl compounds contribute to ozone pollution in the CPUA, particularly through their secondary formation processes. This new understanding helps fill the existing gap in knowledge and provides valuable insights into the sources and chemical pathways of carbonyls in the region. We believe these clarifications strengthen the overall contribution of our study.

Lines 715-731:

" Compared to clean days, carbonyl compound concentrations were significantly higher on ozone pollution days, with increases ranging from 22.8% to 66.2%. Between 19.5% and 48.6% of the total volatile organic compound (VOC) ozone formation potential (OFP) was attributed to the 15 carbonyl compounds, highlighting their substantial contribution to ozone formation, particularly formaldehyde and acetaldehyde. While primary emissions are the main sources of these compounds, secondary formation processes contributed over 30% on average to the concentrations of formaldehyde, acetaldehyde, and acetone. Under ozone pollution conditions, the secondary formation concentrations of these three compounds were notably higher than on clean days, with increases of 58.8%, 54.6%, and 57.6%, respectively, emphasizing the critical role of secondary processes in exacerbating regional ozone pollution. OBM modeling revealed that formaldehyde and acetaldehyde primarily originated from the secondary formation of alkenes and

BVOCs, while acetone mainly stemmed from the secondary formation of alkanes. These findings highlight that while the concentration of carbonyl compounds is important, their significant impact on ozone formation is primarily driven by secondary chemistry. Specifically, the secondary formation of these compounds from alkenes and biogenic BVOCs plays a key role in this process."

**Comment 14:** Line 707. Define FNR again here for those who skip to the conclusions.

**Response:** Thank you for the suggestion. We have added the definition of FNR (the formaldehyde to $NO_2$ ratio) in the revised manuscript to ensure clarity for readers who may skip directly to the conclusions.

---

## Author Response (AR3)

**Response to the Comments of the Reviewers**

**---For the manuscript "egusphere-2024-1204"**

Dear Editor and Reviewers,

We acknowledge the constructive comments and encouragement of the reviewers and are grateful for the efficient service of the editor. Here, we submit our revised manuscript titled "Exploring the Crucial Role of Atmospheric Carbonyl Compounds in Regional Ozone heavy Pollution: Insights from Intensive Field Observations and Observation-based modelling in the Chengdu Plain Urban Agglomeration, China" (Manuscript ID: egusphere-2024-1204), along with a thorough, point-by-point response to each comment raised by the reviewers. The revisions to the manuscript are highlighted in blue text in the attached "Response to the Comments of the Reviewers." Additionally, we have provided a clean version of the revised manuscript as required. We greatly appreciate the reviewers' insightful comments and valuable suggestions, which have significantly improved the quality of our manuscript.

Sincerely yours,

Authors of the manuscript egusphere-2024-1204

Corresponding author: Hong Li (lihong@craes.org.cn)

First author: Jiemeng Bao (2301112284@stu.pku.edu.cn)

Dec. 19, 2024

**Response to the Reviewer #1**

**Comment 1:** Make title compatible with guidelines to be concise and to highlight the findings rather than the topic. For example: Atmospheric Carbonyl Compounds are crucial in Regional Ozone heavy Pollution: Insights from the Chengdu Plain Urban Agglomeration, China.

**Response:**

We sincerely thank the editor for the valuable feedback. Following the recommendation, we have revised the title to: "Atmospheric Carbonyl Compounds Are Crucial in Regional Ozone Heavy Pollution: Insights from the Chengdu Plain Urban Agglomeration, China". This revised title aligns with the guidelines by being concise and highlighting the key findings of our study. Thank you for your guidance.

Lines 1-3:

"Atmospheric Carbonyl Compounds are crucial in Regional Ozone heavy Pollution: Insights from the Chengdu Plain Urban Agglomeration, China"